# Efficient Genome Editing Using ‘NanoMEDIC’ AsCas12a-VLPs Produced with Pol II-Transcribed crRNA

**DOI:** 10.3390/ijms252312768

**Published:** 2024-11-27

**Authors:** Sofiia E. Borovikova, Mikhail V. Shepelev, Dmitriy V. Mazurov, Natalia A. Kruglova

**Affiliations:** 1Institute of Gene Biology Russian Academy of Sciences, 119334 Moscow, Russia; borovikova_sofiya@mail.ru; 2Center for Precision Genome Editing and Genetic Technologies for Biomedicine, Institute of Gene Biology Russian Academy of Sciences, 119334 Moscow, Russia; mshepelev@mail.ru (M.V.S.); dmitrii.mazurov@yale.edu (D.V.M.)

**Keywords:** virus-like particles, CRISPR/Cas delivery, AsCas12a, NanoMEDIC, ribonucleoprotein complexes, Pol II promoter

## Abstract

Virus-like particles (VLPs) are an attractive vehicle for the delivery of Cas nuclease and guide RNA ribonucleoprotein complexes (RNPs). Most VLPs are produced by packaging SpCas9 and its sgRNA, which is expressed from the RNA polymerase III (Pol III)-transcribed U6 promoter. VLPs assemble in the cytoplasm, but U6-driven sgRNA is localized in the nucleus, which hinders the efficient formation and packaging of RNPs into VLPs. In this study, using the nuclease packaging mechanism of ‘NanoMEDIC’ VLPs, we produced VLPs with AsCas12a and exploited its ability to process pre-crRNA. This allowed us to direct crRNA in the cytoplasm as part of a Pol II-driven transcript where AsCas12a excised mature crRNA, thus boosting RNP incorporation into VLPs. CMV-driven crRNA increased *Venus* and *CCR5* transgene knockout levels in 293 cells from 30% to 50–90% and raised the level of endogenous *CXCR4* knockout in Jurkat T cells from 1% to 20%. Changing a single crRNA to an array of three or six identical crRNAs improved *CXCR4* knockout rates by up to 60–70%. Compared to SpCas9-VLPs, the editing efficiencies of AsCas12a-VLPs were higher, regardless of promoter usage. Thus, we showed that AsCas12a and CMV-driven crRNA could be efficiently packaged into VLPs and mediate high levels of gene editing. AsCas12a-VLPs are a new and promising tool for the delivery of RNPs into mammalian cells that will allow efficient target genome editing and may be useful for gene therapy applications.

## 1. Introduction

The CRISPR/Cas technology enables precise genetic manipulations and is a promising tool to treat many human diseases [1]. Its importance for disease correction is demonstrated by dozens of ongoing clinical trials and by the approval of the first CRISPR/Cas9-based cellular gene therapy in December 2023 (clinicaltrials.gov, [2]). Despite the efficiency and specificity of different Cas proteins, which are continually improving, the delivery of genome editors to therapeutically relevant targets remains challenging. The Cas nuclease and guide RNA (gRNA) delivered into mammalian cells as ribonucleoprotein complexes (RNPs) have a compelling advantage over DNA- and RNA-based expression because RNPs are short-lived, and this substantially reduces the risk of off-target DNA damage [3]. Different methods of RNP delivery have been described [4,5], but electroporation remains one of the most effective approaches, which has been widely adopted for RNP delivery into immune and hematopoietic cells ex vivo [6,7]. However, a clear drawback of the electroporation procedure is a high rate of cell mortality induced by electric shock and difficulties in application in vivo. Thus, an alternative approach for safe and efficient RNP delivery into primary cells is very desirable for the clinical usage of CRISPR/Cas.

Virus-like particles (VLPs) represent an attractive vehicle for the delivery of a variety of biomolecules. They can self-assemble in producer cells, incorporate diverse protein or mRNA cargos, can be pseudotyped with a certain Env protein that will mediate transduction of different or specific cell types, and can be used in vivo [8]. Several VLP systems predominantly based on HIV or MLV have been designed to incorporate CRISPR/Cas9 RNPs [9,10,11,12]. VLPs can deliver not only the SpCas9 nuclease but also more complex and large proteins such as base editors [12] or prime editors [13]. However, none of the VLP-based editing systems have exploited nucleases other than SpCas9. The packaging was mostly achieved by fusing the SpCas9 nuclease and Gag polyprotein, exemplified by ‘Nanoblades’ [9], Cas9-VLPs [10], eVLPs [12], and their derivative systems [13,14]. To release SpCas9 from Gag and allow for its migration into the nucleus in target cells, a protease cleavage sequence is introduced between SpCas9 and Gag, which is cleaved in VLPs after particle maturation. Alternatively, ‘NanoMEDIC’ VLPs rely on a noncovalent complex formation between FRB–SpCas9 and FKBP12–Gag in the presence of a rapamycin analog, AP21967 [11]. As these VLPs are produced without a viral protease, they are devoid of the effect of nonspecific SpCas9 cleavage described for the HIV protease that reduces the editing activity of VLPs [11].

High editing levels have been reported for all aforementioned VLP systems tested in different human cell lines and primary cells, and some successful results have been obtained in in vivo mouse models [9,10,11,12,13]. Another approach of recruiting RNPs into VLPs via gRNA has also been described, but the gene editing efficiency of these particles was low [15].

The editing efficiency of VLPs depends on two main factors: the efficiency of VLP assembly with modified Gag proteins and the efficiency of RNP incorporation into particles. A higher ratio of unmodified to modified Gag molecules supports VLP assembly but reduces SpCas9 packaging, and this parameter could be manipulated by changing the amount of transfected plasmids as shown for Gag-SpCas9 and Gag-Pol plasmid ratios [10,12]. Therefore, it is important to find the optimal ratio of transfected plasmid quantities for VLP production. RNP packaging requires that both the nuclease and gRNA be efficiently incorporated into VLPs. In most systems, only the SpCas9 nuclease is specifically directed into particles, whereas gRNA is packaged via its association with the nuclease [9,10,11,12,13]. One VLP system uses the opposite approach with MS2-coupled gRNA-specific incorporation and the concomitant passive packaging of nuclease [15]. Finally, specific packaging for both the nuclease and gRNA has been suggested [13].

All VLP systems except for ‘NanoMEDIC’ VLPs utilize gRNA, which is expressed from the RNA polymerase III (Pol III)-transcribed U6 promoter. U6-driven gRNA is localized in the nucleus, while VLPs assemble in the cytoplasm. This suboptimal gRNA localization hinders the formation and packaging of RNPs into VLPs and represents an important limiting factor for efficient genome editing using VLPs. An insufficient level of gRNA in the particles was indirectly confirmed by Gee et al. and Hamilton et al., who showed that transient transfection and expression of gRNA in target cells or the addition of the U6-gRNA cassette into an HIV reporter increased the editing efficiency of ‘NanoMEDIC’ VLPs [11] or Cas9-VLPs [10], respectively. Both observations suggest that a fraction of SpCas9 molecules in VLPs were not bound to gRNA. To improve gRNA packaging into VLPs, Hamilton et al. inserted a U6-gRNA cassette in all plasmids used for Cas9-VLP production, which substantially improved VLP editing efficiency [14]. For eVLP production, An et al. combined Gag-SpCas9 incorporation and MS2-gRNA packaging, which showed an additive effect [13].

An alternative approach was proposed by Gee et al. [11]. The authors hypothesized that gRNA expressed from the U6 promoter and localized in the nucleus is not available for packaging and must be transported to the cytoplasm where VLP assembly takes place. To test this, gRNA was flanked by ribozymes and expressed from an RNA polymerase II promoter (Pol II). However, the self-cleavage activity of ribozymes is not limited to the cytoplasmic compartment, and the efficiency of gRNA excision by ribozymes was not validated directly; therefore, the contribution of this mechanism to VLP editing remains unclear [11].

We sought to improve gRNA packaging into VLPs by localizing gRNAs into the cytoplasm and chose the ‘NanoMEDIC’ system for optimization. To direct gRNA into the cytoplasm, it should be first included in a Pol-II transcript and then liberated from it. To achieve this, several approaches have been developed including flanking gRNA with ribozymes, tRNA sequences, or Csy4 endonuclease motifs [11,16,17]. All of them demonstrated either low efficiency or required the expression of additional enzymes, which is highly undesirable for VLP production. A possible good alternative to the aforementioned techniques is to use the AsCas12a nuclease, which can interact with the cognate pre-crRNA, process it to crRNA, and form an active RNP complex [18,19]. In contrast to SpCas9 which exploits a two-component gRNA consisting of crRNA and tracrRNA, which have to be produced by enzymes other than SpCas9 and can be combined into a single sgRNA molecule of about 100 nt, AsCas12a uses a crRNA molecule of about 40 nt and can excise it from a long pre-crRNA transcript due to its RNA cleavage activity [18] (for simplicity, we use the following names throughout the manuscript: crRNA for AsCas12a, sgRNA for SpCas9, and gRNA for both nucleases). AsCas12a requires a T-rich PAM (TTTV) in the 5′-end of the protospacer and produces sticky 5′-ends as compared to SpCas9, which recognizes the NGG PAM in the 3′-end of the protospacer and leaves blunt ends [20,21]. Importantly, unlike SpCas9, the 5′-end localization of AsCas12a PAM enables multiple cycles of editing. The different PAM requirement for AsCas12a allows for expanding the number of targeted loci, whereas sticky DNA ends can potentially boost the repair of double-stranded breaks by the homology-directed repair pathway [20]. Additionally, AsCas12a was found to have higher specificity than SpCas9 [22]. Several studies reported efficient gene editing with AsCas12a and crRNA expressed from a single Pol II-driven construct, including successful multiplexed editing, but this nuclease has not been adapted for VLP incorporation [19,23]. We hypothesized that we could use the pre-crRNA cleavage activity of AsCas12a to direct crRNA into the cytoplasm, where it would be excised by AsCas12a followed by RNP formation and packaging into particles, thus improving VLP editing efficiency.

In this study, we produced VLPs based on the packaging mechanism of ‘Nano-MEDIC’ particles but incorporated the AsCas12a nuclease instead of the SpCas9 that was used in all previous VLP designs. To increase RNP packaging, we directed crRNA into the cytoplasm by expressing it as part of a Pol II-driven transcript under the control of a CMV promoter. As a result, using AsCas12a-VLPs, we managed to increase editing efficiency 3–20-fold depending on the locus and target cells. We compared VLPs with the AsCas12a or SpCas9 nuclease produced with either U6- or CMV-driven gRNA and found that AsCas12a-VLPs demonstrated superior editing efficiency, regardless of the promoter used. AsCas12a-VLPs with CMV-driven crRNA efficiently targeted HIV coreceptor genes and enabled editing in Jurkat T lymphocytes, which was not possible using similar VLPs with U6-driven crRNA or for VLPs with SpCas9.

## 2. Results

### 2.1. Guide RNA Selection for Target Loci

For validation of AsCas12a-VLPs, we selected editing targets in three loci: HIV coreceptor genes *CXCR4* and *CCR5*, which are promising targets for HIV gene therapy, and the *Venus* reporter stably integrated into the genome of the HEK293 cell line. We have previously shown efficient editing of an endogenous *CXCR4* locus in the CEM/CCR5 cell line and in primary CD4+ T cells, including the knock-in of therapeutically relevant constructs into this locus [24]. In contrast to CXCR4, the CCR5 receptor is expressed in CEM/CCR5 cells from an integrated lentiviral construct; its level in the population is low and heterogeneous. Therefore, *CCR5* knockout was assessed in CEM/CCR5 clone #8 cells that we selected for their high CCR5 surface level (Appendix A). The *Venus* reporter cell line was generated based on the previously described Traffic Light Reporter 5 (TLR5) system [25] via knock-in of the Venus repair donor construct followed by sorting of Venus+/RFP− cells (Appendix A).

We selected several previously characterized targets for SpCas9 and AsCas12a in the *CXCR4* and *CCR5* genes. Three AsCas12a targets were selected in the *CXCR4* and *CCR5* loci [26,27,28,29], two targets were selected for the SpCas9 nuclease in the *CCR5* locus [30,31], and one target for the *CXCR4* locus [24]. To edit the *Venus* reporter, we selected one target for SpCas9 and one target for AsCas12a, which are located in the same site of the *Venus* cDNA and are characterized in the study by Xin et al. [32]. The positions of editing targets in selected loci are shown in Figure 1A.

Next, we generated the 3xHA-tagged AsCas12a expression plasmid encoding henAsCas12a-HF (hereafter referred to as AsCas12a) based on the plasmid described in Kleinstiver et al. [33], as well as plasmids for the expression of selected gRNAs, and assessed the editing efficiency in cells via electroporating CEM/CCR5 cells or transfecting Venus reporter cells with a combination of plasmids coding for a CMV promoter-driven nuclease and U6 promoter-driven gRNA.

Editing efficiency (*Venus* knockout level) in clone #8 of Venus reporter cells that have a single copy of the transgene in the genome was measured six days after transfection by flow cytometry and was found to be comparable for both nucleases (Figure 1B, Appendix A).

The editing efficiency of endogenous *CXCR4* and lentivirus-encoded *CCR5* was measured by flow cytometry 5 days after electroporation in a pool of CEM/CCR5 cells or CEM/CCR5 clone #8 cells, respectively (Figure 1C,D). We found that all gRNAs allowed for the efficient editing of their respective targets, except crRNA#1 for *CXCR4*. To simplify the subsequent comparison of VLPs with SpCas9 and AsCas12a, we chose gRNAs that ensured similar knockout levels for both nucleases. Specifically, crRNA#3 was selected for the *CXCR4* locus, whereas sgRNA#2 and crRNA#2 were chosen for the *CCR5* locus.

### 2.2. Generation of AsCas12a-VLPs

To produce AsCas12a-VLPs, we generated plasmids encoding 3xHA-tagged AsCas12a fused with the FRB domain at the N- or C-terminus and evaluated the editing efficiency of these fusion proteins (Figure 2A). We found that the level of FRB-AsCas12a in transfected 293T cells was similar to that for the unmodified nuclease, whereas the level of AsCas12a-FRB was much lower (Figure 2B). The level of the *CXCR4* knockout was then tested by electroporating these plasmids into CEM/CCR5 cells together with the plasmid encoding crRNA#3 against the *CXCR4* gene (Figure 2C). Both variants of FRB-fused nucleases induced lower knockout levels as compared to unmodified AsCas12a (77% for FRB-AsCas12a and 62% for AsCas12a-FRB). The lower editing efficiency of AsCas12a-FRB is likely due to the reduced level of this fusion protein in cells. Based on these results, we selected the FRB-AsCas12a fusion variant for further experiments.

Next, we produced VLPs with the FRB-AsCas12a nuclease in parallel with ‘NanoMEDIC’ particles assembled using original plasmids encoding FRB-SpCas9 and FKBP12-Gag [11]. We replaced the original plasmid for sgRNA expression under the control of the LTR promoter with pKS-based plasmids encoding gRNA under the control of the U6 promoter, since the original plasmid contained ribozymes, whose efficiency in excising sgRNA was not evaluated [11]. Moreover, this plasmid outperformed the plasmid coding for U6-driven sgRNA only in the presence of an exogenously expressed Tat protein [11].

First, we evaluated the protocol for the production of VLPs with SpCas9 and compared two methods of VLP concentration. Plasmids for VLP assembly (pHLS-FRB-SpCas9, pHLS-FKBP12-Gag, pCMV-VSVG, and pKS-U6-sgRNA) were transfected into 293T cells, which were incubated in the presence of a rapamycin analog, AP21967, for 48 h. VLPs were harvested and concentrated approximately 50-fold by centrifugation at 21,000× *g* for 2.5 h or by ultracentrifugation on top of 20% (*w*/*v*) sucrose in PBS at 112,400× *g* for 2 h [10]. Concentration conditions used by Gee et al. for the original ‘NanoMEDIC’ VLP production included prolonged overnight centrifugation [11] and were not tested in our study. VLP samples were brought to the same volume and their editing efficiency was analyzed by *Venus* knockout in 293-Venus clone #8 cells. We found that centrifugation at 21,000× *g* for 2.5 h ensured higher editing efficiency of VLPs as compared with ultracentrifugation, and therefore used this protocol in all subsequent experiments (Appendix A).

Using the optimized protocol, we produced VLPs containing SpCas9 and AsCas12a to target the *Venus* reporter (Figure 2D) and analyzed nuclease packaging into particles, which was comparable for both nucleases (Figure 2E).

To evaluate the input of gRNAs in the editing efficiency of VLPs, one day before transduction, target cells were transfected with the plasmid encoding sgRNA/crRNA (Figure 2F, (+) samples) or with the plasmid encoding a corresponding irrelevant gRNA (Figure 2F, (−) samples). The next day, cells were harvested with trypsin, and 4.5 × 10^4^ cells were transduced in suspension with two doses of VLPs in a total volume of 90 µL, since we found that cells in suspension were transduced better than in a monolayer (Appendix A).

On day 6 after transduction, *Venus* knockout levels were measured by flow cytometry. The mean knockout level for the 50 μL dose was 40% for VLPs with AsCas12a and 21% for VLPs with SpCas9 (Figure 2F, (−) samples). The effect of VLPs was dose-dependent, and pre-transfection with the gRNA-expressing plasmid slightly increased knockout levels ((+) vs. (−) samples).

Next, VLPs targeting the *CXCR4* and *CCR5* genes were produced, and their editing activity was tested by flow cytometry in CEM/CCR5 or CEM/CCR5 clone #8 cells, respectively. Unfortunately, we were unable to detect any level of VLP-mediated *CXCR4* and *CCR5* editing. Since CEM/CCR5 cells are efficiently transduced with VSVG pseudotyped lentiviral particles [24], we believe that the lack of editing is explained by some cellular factors that limit RNP disassembly from VLPs. Therefore, as another model for VLP-mediated editing, we used 293T/CD4/CCR5 clone #19 cells, stably expressing CD4 and CCR5 at a high level (Appendix A—flow cytometry analysis of coreceptor levels on the 293T/CD4/CCR5 clone #19 cells), which have been previously described [24].

The experiment was conducted in a way similar to that described for *Venus*, with a comparable content of SpCas9 and AsCas12a in VLPs (Appendix A), and showed even higher editing levels for *CCR5*. The mean knockout level with the 50 μL dose was 56% for VLPs with AsCas12a and 34% for VLPs with SpCas9 (Appendix A). In the case of gRNA pre-transfection, mean knockout levels reached 69% for AsCas12a and 55% for SpCas9 (Appendix A, (+) vs. (−) samples).

Taken together, we showed that AsCas12a is efficiently packaged into VLPs and mediates higher knockout levels of *Venus* and *CCR5* in stably transfected HEK293 target cells in comparison to ‘NanoMEDIC’ VLPs with SpCas9. Pre-transfection of gRNA-expressing plasmids into the target cells increased the editing efficiency for both nucleases, suggesting that the packaging of gRNA into VLPs is suboptimal, and a fraction of nuclease molecules in VLPs are not associated with gRNA. This is consistent with previous studies in which RNP was incorporated into VLPs as a SpCas9 fusion with Gag [10,11]. Our finding suggests that it is possible to increase the level of active RNP molecules in VLPs by increasing the efficiency of gRNA packaging.

### 2.3. Genome Editing with AsCas12a and Pol II Promoter-Driven crRNA

When expressed under the control of the U6 promoter used by Pol III, sgRNA accumulates in the nucleus, whereas VLP assembly occurs in the cytoplasm beneath the plasma membrane. Hence, to increase the efficiency of sgRNA packaging into ‘NanoMEDIC’ VLPs, Gee et al. proposed to express SpCas9 sgRNA as part of the LTR promoter-driven transcript of Pol II, thus directing sgRNA into the cytoplasm where its excision from the transcript is performed by flanking ribozymes [11]. However, improved editing efficiency of VLPs with the ribozyme-containing transcript was observed only in the presence of the LTR promoter, the Ψ packaging signal, and the overexpression of Tat. When the Ψ signal was omitted, the editing levels of VLPs were even lower than those for VLPs produced with U6-driven sgRNA, which suggests that ribozymes cannot efficiently excise sgRNA [11]. In contrast to SpCas9, which requires complicated and largely ineffective mechanisms to liberate sgRNA from the transcript of Pol II, AsCas12a has intrinsic pre-crRNA processing activity, which makes it easy to create constructs for the expression of crRNA from the Pol II promoter [19]. Therefore, we decided to use this feature of AsCas12a to direct crRNA into the cytoplasm and thus improve its packaging into VLPs.

First, we generated constructs expressing the reporter mClover under the Pol II-controlled CMV promoter and placed the sequence of crRNA#3 (cr1X4) for the *CXCR4* gene in the 3′-end part of the transcript (pCMV-mClover-trpl-DR-cr1X4-DR; Figure 3A). The *CXCR4* target was chosen as an endogenous locus to test our hypothesis in a more relevant model in CEM/CCR5 cells, as compared to *Venus* and *CCR5* transgenes. To allow for the excision of crRNA by AsCas12a, we flanked it by two direct repeats (DRs). We also produced constructs with or without the Triplex sequence from *Malat1* long noncoding RNA that is known to protect transcripts lacking the polyA tail from degradation after AsCas12a-mediated excision of crRNA, as was shown by Campa et al. [23].

The editing efficiency of generated plasmids was evaluated in CEM/CCR5 cells by *CXCR4* knockout. In comparison to the control U6 promoter-driven crRNA, CMV promoter-transcribed crRNA mediated approximately 2-fold lower knockout levels with slightly higher values for the Triplex-containing transcript (Figure 3B). However, the level of *CXCR4* knockout could be easily increased by raising the dose of the pCMV-mClover-trpl-DR-cr1X4-DR plasmid to reach the level of knockout produced with the pKS-U6-crRNA plasmid (Figure 3C).

Importantly, the AsCas12a variant H800A, which is deficient for RNA cleavage activity, did not induce *CXCR4* knockout when combined with the CMV promoter-driven crRNA but was similarly active to non-mutated AsCas12a when combined with U6-driven crRNA (Figure 3B). The levels of both proteins in the lysates of 293T cells were similar (Appendix A). This result indicates that genome editing with AsCas12a crRNA expressed under the control of the Pol II promoter is dependent on the cleavage of the crRNA-containing transcript by AsCas12a. We obtained additional indirect evidence confirming AsCas12a-mediated transcript cleavage by comparing the mean fluorescence intensity (MFI) values of mClover in samples with or without Triplex (Appendix A). In the case of non-mutated AsCas12a, 5 days after electroporation, MFI values were significantly reduced for samples without Triplex, suggesting that the transcript was susceptible to degradation after the loss of the polyA tail. In the case of AsCas12a H800A, the overall MFI values were higher, reflecting only partial transcript protection by Triplex, yet no difference was observed between samples with or without Triplex.

Additionally, we tested the editing efficiency of AsCas12a in combination with Pol II promoter-expressed crRNA for the *CCR5* and *Venus* loci. For both targets, the knockout level with CMV-driven crRNA expression was comparable to that achieved with U6-driven crRNA expression. *CCR5* knockout levels increased with the increasing dose of crRNA-expressing plasmid (Figure 3D). The level of the *Venus* knockout was not further increased by the amount of plasmid, which is possibly explained by the saturating amount of transfected DNA (Figure 3E). Control plasmids with irrelevant crRNA did not induce editing for both U6- and CMV-driven crRNA expression (Appendix A).

In the original study on ‘NanoMEDIC’ by Gee et al., the authors also suggested expressing sgRNA as part of the Pol II-dependent transcript and excising it by ribozymes [11]. We sought to compare the crRNA processing efficiency of AsCas12a with sgRNA excision by ribozymes as described by Gee et al. [11]. For this purpose, we generated a plasmid of a similar design to pCMV-mClover-trpl-DR-cr1X4-DR, in which the sgRNA for SpCas9 targeting *CXCR4* was flanked by HH and HDV ribozymes (Figure 3F).

The efficiency of cleavage was determined indirectly by the level of *CXCR4* knockout in CEM/ССR5 cells. For comparison, we included samples with gRNA expressed under the control of the U6 promoter. Similar to the results shown in Figure 3B, CMV promoter-driven crRNA mediated ~50% *CXCR4* knockout relative to U6-driven crRNA (Figure 3G). At the same time, CMV-driven SpCas9 sgRNA transfected at the same dose did not induce editing (Figure 3G). Only a three-fold increase in the amount of this plasmid led to a reasonable knockout, which was still significantly lower than that of CMV promoter-driven crRNA and constituted only 21% of the knockout level with U6-driven sgRNA expression. These data suggest that ribozyme-mediated excision of SpCas9 sgRNA is inefficient and that AsCas12a, with its straightforward mechanism of crRNA excision from Pol II transcripts and subsequent cytoplasm localization, may be advantageous in comparison to SpCas9 for VLP production.

### 2.4. AsCas12a and crRNA Can Be Expressed from a Single Pol II-Driven Transcript That Is Compatible with the ‘NanoMEDIC’ VLP System

Since efficient editing was shown for CMV promoter-driven crRNA (Section 2.3), we hypothesized that a single construct encoding both AsCas12a and crRNA under the control of the CMV promoter would be advantageous for VLP production due to a reduced number of transfected plasmids. To this end, we replaced the mClover cDNA with the AsCas12a sequence in the pCMV-mClover-trpl-DR-cr1X4-DR plasmid (Figure 4A, construct #2) to produce a plasmid encoding AsCas12a and crRNA, which is later referred to as a single plasmid. We analyzed the *CXCR4* knockout level in CEM/CCR5 cells induced by this plasmid in comparison to a combination of two separate plasmids encoding CMV promoter-driven AsCas12a and U6 promoter-driven crRNA (Figure 4A, construct #1). The single plasmid (construct #2) induced three-fold lower *CXCR4* editing, as compared to the combination of plasmids (10% vs. 30%) (Appendix A), when electroporated at the same amount (0.6 pmol) as an AsCas12a-encoding plasmid shown in Figure 4A (construct #1). To improve the editing level, we increased the amount of the single plasmid to 0.9 pmol. Under these conditions, the level of the *CXCR4* knockout in cells electroporated with the single plasmid was 20%, compared to 35% when two plasmids were used (Figure 4A, bars #1 and #2). Based on these data, we used 0.9 pmol of the single plasmid in subsequent experiments.

mRNA transcribed from the single plasmid may produce only one molecule of crRNA upon its excision by AsCas12a and multiple molecules of the AsCas12a nuclease due to the presence of the Triplex sequence protecting the transcript from degradation. Thus, the single plasmid provides a suboptimal molar ratio of AsCas12a to crRNA. The deficit in crRNA relative to AsCas12a molecules may lead to inefficient editing. Therefore, we sought to increase the editing efficiency when using a single construct by optimizing its 3′UTR, which contains crRNA. To achieve this, we tested three different approaches. First, we increased the copy number of the spacer and produced plasmids with arrays of two or three identical spacers flanked by direct repeats (Figure 4A, constructs #3 and #4), similarly to plasmids used for multiplex editing with AsCas12a [23]. Second, we modified the sequence of direct repeats by adding four T nucleotides onto its 3′-end (Figure 4A, constructs #5, #6, and #7), which is expected to boost the cleavage efficiency of direct repeats by AsCas12a, as was shown by Magnusson et al. [34]. Finally, we hypothesized that the Triplex sequence placed next to the direct repeat may impede its cleavage by AsCas12a due to its folding into a complex 3D structure [35]. Hence, the distance between Triplex and the direct repeat must be increased to enable efficient crRNA processing. To test this hypothesis, we generated constructs with 50 or 150 nucleotides instead of the original five nucleotides placed between Triplex and the direct repeat. Insert sequences represented noncoding fragments from different plasmids (for details, see Materials and methods) that were not predicted to fold into complex structures (Figure 4A, constructs #8, #9, and #10).

Next, the *CXCR4* knockout levels in CEM/CCR5 cells were analyzed with all obtained plasmids. Among all three types of modifications, only the constructs with two or three copies of the spacer sequence increased the level of knockout to 27% independently of the sequence of the direct repeat (constructs #3, #4, #6, and #7, Figure 4B). No difference was observed between plasmids with two or three spacers. Modification of the direct repeat or increased distance between Triplex and the direct repeat did not affect editing.

Taken together, we found that a single construct for the expression of AsCas12a and crRNA under the control of the CMV promoter allows for the achievement of about 80% of the *CXCR4* knockout level when using an increased plasmid amount and 2–3 copies of spacers as compared to using two separate plasmids. Therefore, we decided to adapt it to the ‘NanoMEDIC’ VLP production system by adding the FRB domain to the N-terminus of AsCas12a (Figure 4C, construct #4). For the single plasmid, the level of the *CXCR4* knockout was not affected by the presence of the FRB domain (Figure 4D, constructs #3 and #4) and could be increased to levels similar to those obtained with a combination of two separate plasmids (constructs #1 and #2) by increasing the amount of the electroporated plasmid to 1.2 pmol (Figure 4D).

Single plasmids adapted to the ‘NanoMEDIC’ system were then produced for *CCR5* and *Venus* loci and tested in 293T/CD4/CCR5 clone #19 or 293-Venus clone #8 cells, respectively (Appendix A). The editing levels for both constructs were comparable to those without FRB (Appendix A).

Western blotting revealed that single plasmids produced lower levels of AsCas12a or FRB-AsCas12a compared to separate plasmids encoding the corresponding protein (Appendix A). Given the similar editing efficiency of separate plasmids and the single construct at a high dose, we hypothesize that the activity of AsCas12a expressed from a single plasmid is limited predominantly by the availability of free crRNA.

We showed that two or three copies of crRNA increased *CXCR4* knockout levels when compared with one copy (Figure 4B). Therefore, we decided to test whether multiple copies of crRNA would further increase editing efficiency. To this end, we chose the pCMV-mClover-trpl-DR-cr1X4-DR plasmid, suggesting that using a separate plasmid for crRNA expression would allow for easier adjustment of the dose to ensure an optimal AsCas12a/crRNA ratio. Based on this construct, we generated plasmids containing three and six copies of the spacers for *CXCR4* or *CCR5* and assessed their editing efficiency through *CXCR4* and *CCR5* knockout levels in CEM/CCR5 clone #8 cells (Figure 5A). The plasmids with three or six identical spacers induced a 30% higher knockout level for *CXCR4* compared to the plasmid with a single spacer, and no difference was observed between three and six spacers (Figure 5B). The levels of *CCR5* knockout were similar for plasmids containing one, three, or six identical spacers (Figure 5C). This result suggests that the effect of an increased spacer copy number may vary between target loci. Additionally, *CCR5* is a transgene, and its editing may be efficient even with lower levels of crRNA reaching a plateau at chosen plasmid doses.

The ability of AsCas12a to liberate its own crRNA from transcripts is widely used for multiplexed editing [19,23]. To test the performance of pCMV-mClover-based plasmids in this application, we generated a plasmid coding for six crRNAs: three for *CXCR4* and three for *CCR5* as an array of alternating crRNAs (Figure 5A, construct #9). This plasmid induced *CXCR4* and *CCR5* editing with an efficiency similar to the corresponding plasmids with a single spacer (Figure 5B,C).

Overall, these results indicate that AsCas12a and crRNA expressed as a single transcript of Pol II allow efficient genome editing and can be adapted to the ‘NanoMEDIC’ VLP system. Pol II-driven crRNA is compatible with multiplexing, which can be advantageous for VLP production.

### 2.5. Generation of AsCas12a-VLPs with a CMV Promoter-Driven crRNA

Next, we produced four different types of AsCas12a-VLPs targeting *Venus* or *CCR5* using the optimized protocol. 293T cells were transfected with the pHLS-FKBP12-Gag and pCMV-VSVG plasmids, together with varying combinations of nuclease/crRNA expression plasmids: CMV-AsCas12a + U6-crRNA, which served as a control without the FRB domain required for packaging into VLPs (Figure 6A, sample #1); CMV-FRB-AsCas12a + U6-crRNA (Figure 6A, sample #2); CMV-FRB-AsCas12a + CMV-mClover-crRNA (Figure 6A, sample #3); and CMV-FRB-AsCas12a-crRNA (Figure 6A, sample #4).

VLPs #2 with U6-crRNA, representing the original version of AsCas12a-VLPs (Figure 2), demonstrated knockout levels of around 30% for both *Venus* and *CCR5* (Figure 6B,D, Appendix A). VLPs #4 produced with the single plasmid CMV-FRB-AsCas12a-crRNA induced 1.5–2-fold higher editing rates, with mean values reaching 46% for *Venus* and 61% for *CCR5*. The highest editing levels were achieved with VLPs #3 produced with CMV-FRB-AsCas12a and CMV-mClover-crRNA, which were 2–3-fold higher than those obtained with VLPs #2. The maximal knockout levels reached 67% for *Venus* and 91% for *CCR5* when VLPs were used at the highest dose. Importantly, VLPs #3 induced higher editing levels of *Venus* and *CCR5* at all doses and were even 9-fold more efficient than VLPs #2 with U6-crRNA at the lowest dose. Taken together, AsCas12a-VLPs produced with CMV-driven crRNA transcribed from both the separate plasmid CMV-mClover-crRNA and the single plasmid CMV-FRB-AsCas12a-crRNA outperformed VLPs produced with U6-driven crRNA.

According to the Western blot data, VLP preparations #2 and #3, which differed only in the type of crRNA-expressing plasmid (U6 vs. CMV promoter, respectively), contained similar levels of AsCas12a (Figure 6C,E). Therefore, the observed 2–3-fold higher editing efficiency of VLPs #3 could be attributed to improved packaging of CMV-driven crRNA. VLPs #4, produced with the single plasmid CMV-FRB-AsCas12a-crRNA, did not differ substantially from the best-performing VLPs #3 in nuclease content (Figure 6C,E). Hence, the lower editing efficiency of VLPs #4 in comparison to VLPs #3 could be explained by a suboptimal ratio of AsCas12a to crRNA molecules produced by the single plasmid, thus reducing the amount of AsCas12a molecules bound to crRNA (see Section 2.4).

Unexpectedly, the control VLPs #1, containing AsCas12a without the FRB domain, demonstrated 20% knockout levels for both *Venus* and *CCR5* that were comparable to those produced with VLPs #2 (Figure 6B,D). Western blot of lysates obtained from 293T producer cells and VLP samples showed enrichment of FRB-AsCas12a in VLP preparations #2, #3, and #4 as compared to AsCas12a in VLPs #1 (Figure 6C,E). These data indicate that AsCas12a without the FRB domain is stochastically packaged into VLPs independently of FRB-FKBP12 dimerization, thus resulting in reasonable editing levels (Figure 6B,D).

Given the high editing efficiency of control VLPs #1 containing AsCas12a without the FRB domain (Figure 6B,D, sample #1), we produced additional control VLP preparations for both U6- and CMV-driven crRNA and tested them by *Venus* knockout in 293-Venus clone #8 cells (Appendix A). VLPs with irrelevant crRNA against *CXCR4* did not induce *Venus* editing (Appendix A, samples #2 and #6), whereas VLPs produced without AP21967 (samples #3 and #7) or with AsCas12a lacking the FRB domain (samples #4 and #8) demonstrated only two-fold lower knockout levels compared to the corresponding fully functional VLPs (Appendix A). Accordingly, the Western blot analysis showed only slightly lower AsCas12a levels in VLPs produced without AP21967 or with AsCas12a lacking the FRB domain (Appendix A). This result again indicates that there is a high level of stochastic AsCas12a packaging into VLPs, which is enough to achieve reasonable editing levels in 293 cells with both U6- and CMV-driven crRNA.

Next, we compared the editing levels of the same four VLP preparations, #1–#4, added to target cells with or without preliminary transfection of pKS-U6-crRNA (Supplementary Figure 6A). For *CCR5* knockout, an increase in the editing levels due to crRNA pre-transfection was more pronounced in the samples with U6-driven crRNA (VLPs #1 and #2) than in the samples with CMV-driven crRNA (VLPs #3 and #4), suggesting that more crRNA was incorporated into VLPs in the latter case (Appendix A). For *Venus* knockout, this effect was modest (Appendix A).

Collectively, these results suggest that CMV-driven crRNA expression in producer cells improves the VLP editing efficiency of the *Venus* and *CCR5* transgene loci as compared to classical U6-driven expression. This effect can be explained by the localization of crRNA molecules in the cytoplasm, where they are available for association with AsCas12a and packaging into VLPs.

### 2.6. AsCas12a-VLPs Produced with Pol II-Driven crRNA Allow Efficient Genome Editing in Jurkat T Lymphocytes

Encouraged by improved VLP design with CMV-driven crRNA expression, we repeated our experiments on the transduction of suspension T cell lines. Since the transduction of CEM/CCR5 cells was not successful, we chose a Jurkat T cell line which allows us to achieve *CXCR4* knockout levels comparable to those in CEM/CCR5 when edited with plasmids (Appendix A). We produced four VLP preparations targeting the *CXCR4* locus with the same plasmid combinations described in Section 2.5 for *Venus* and *CCR5* (Figure 7A). A reasonable editing level, reaching 20% *CXCR4* knockout, was observed only for VLPs #3 produced with CMV-FRB-AsCas12a and CMV-mClover-crRNA (Figure 7B, Appendix A). VLPs #4 produced with the single plasmid CMV-FRB-AsCas12a-crRNA were 10-fold less efficient, allowing only a 2% knockout level. VLPs #2 produced with U6-driven crRNA and VLPs #1 produced with AsCas12a lacking the FRB domain and U6-crRNA induced only about 1% *CXCR4* editing.

We then produced three control VLP preparations based on the design of VLPs #3, similar to the control VLPs described above for 293-Venus clone #8 cells. The first control VLP preparation contained an irrelevant crRNA against *Venus*, the second VLPs were produced without AP21967, and the third VLPs were packaged with AsCas12a lacking the FRB domain. None of the controls induced reasonable levels of *CXCR4* editing (Appendix A). The highest level of 4% for VLPs bearing AsCas12a without FRB was 6-fold lower than the level of 24% obtained with the fully functional VLPs #3. The Western blot analysis of the control VLPs showed only slightly reduced nuclease content in VLPs produced without AP21967 or with AsCas12a lacking FRB (Appendix A), reflecting high levels of non-specific packaging similar to the data presented above for VLPs targeting *Venus* and *CCR5* (Figure 6C,E). In contrast to the results obtained for 293 cells, *CXCR4* knockout levels in Jurkat T cells were more than 6-fold lower for control VLPs (‘-AP21967′ and ‘-FRB’), compared to fully functional VLPs, which suggests that a higher level of RNPs in particles is required for the efficient editing of difficult-to-transduce Jurkat T cells.

Next, we used VLP preparation #3 to compare both SpCas9 and AsCas12a with CMV-driven gRNA expression. In accordance with the data shown above for the ribozyme-containing transcript of sgRNA (Figure 3G), no editing was observed for SpCas9-VLPs (Appendix A). However, in both cases, Jurkat T cells were successfully transduced, which was evident by the weak signal of the mClover reporter passively packaged into particles (Appendix A).

The editing efficiency of VLPs #3 was 1.5–2-fold lower than the efficiency of RNP electroporation (Figure 7C). This indicated that further improvements to VLP design could increase editing levels. One approach was to increase the amount of the plasmid encoding crRNA in a transfection mixture. We produced the best-performing type of VLPs #3 with a dose of 1.66 μg or 4.98 μg of the crRNA plasmid and found that the tripled amount of the crRNA plasmid increased the level of *CXCR4* knockout in Jurkat T cells from 27% to 40% (Figure 7D).

As a second approach to improve VLP editing efficiency, we replaced the plasmid pCMV-mClover-trpl-DR-cr1X4-DR with the crRNA-expressing plasmid coding for three or six identical spacers and found that this led to a 3-fold increase in *CXCR4* knockout levels, reaching 60% and 70% for the plasmids with three and six spacers, respectively (Figure 7E). This effect was even more pronounced at lower VLP doses. A VLP dose of 6 μL induced 20% and 30% *CXCR4* knockout for the plasmids with three and six spacers, respectively, whereas only 1–2% *CXCR4* knockout was observed for the crRNA plasmid with one spacer. Interestingly, unlike VLPs, plasmids with several copies of the spacer only slightly improved editing levels when delivered into target cells via electroporation (Figure 5B). This suggests that increasing the spacer copy number in Pol II-driven crRNA transcripts minimally affects already saturating editing levels when AsCas12a and crRNA are expressed in target cells. However, it becomes crucial for VLP production presumably by increasing the level of crRNA in the cytoplasm.

A third approach to increasing VLP editing efficiency was to improve the transduction of lymphoid cells by pseudotyping VLPs with a combination of VSVG and baboon endogenous retrovirus Rless glycoprotein (BaEVRless), as was shown for ‘Nanoblades’ VLPs [36]. To evaluate how the BaEVRless envelope affects the editing efficiency of ‘NanoMEDIC’ VLPs and whether it is advantageous for the transduction of Jurkat T cells, we produced VLPs #3 coated with VSVG or VSVG+BaEVRless. The dual-envelope coating of VLPs ensured a 1.5-fold increase in *CXCR4* knockout levels (Figure 7F). When the VSVG+BaEVRless coating was combined with the crRNA expression plasmid pCMV-mClover-trpl-DR-cr6X4-DR coding for six spacers, *CXCR4* knockout levels increased almost 4-fold from 20% to 77% for the 50 μL dose and 16-fold from 3% to 48% for the 6 μL dose (Figure 7E), reflecting an additive effect of these two approaches.

Next, we transduced primary CD4+ T cells from two donors with VLPs #3 produced with different crRNA-expressing plasmids (Appendix A). We used pCMV-mClover-trpl-DR-cr1X4-DR at a dose of 1.66 μg, the tripled dose of this plasmid (crRNA plasmid, μg ×3), or the plasmid coding for three spacers (crRNA spacer ×3) at a dose of 1.66 μg. The pCMV-eBFP2-trpl-DR-cr1Venus-DR plasmid used at the same dose was chosen as a control. On day 4 after transduction, surface CXCR4 levels were analyzed by flow cytometry. Control VLPs targeting *Venus* reduced the number of CXCR4-positive cells by 2–5% relative to non-transduced cells, reflecting the influence of the VLP transduction procedure on the cell state (Appendix A). Therefore, the *CXCR4* knockout levels were calculated by subtracting a fraction of CXCR4-negative cells for control VLPs from that in the samples with crRNA against *CXCR4*. For the first donor, using the tripled amount of the crRNA plasmid or the plasmid with three spacers increased editing levels from 2% to 10%, whereas for the second donor, there was no difference between the three VLP types, with an editing level of around 8% (Appendix A).

Collectively, the results obtained for Jurkat T cells indicate that CMV-driven crRNA expression in VLP producer cells not only improves AsCas12a-VLP editing efficiency, as was shown in the previous section for the 293 cells, but is required to achieve reasonable knockout levels for cells that are difficult to transduce. Editing efficiency can be further improved by increasing the amount of crRNA in producer cells during VLP production, either with an increased amount of the crRNA-expressing plasmid or more substantially by using a Pol II transcript with an array of multiple copies of crRNA. Finally, editing levels can be increased by enhancing transduction efficiency, as exemplified by the use of a dual-envelope coating with VSVG and BaEVRless.

## 3. Discussion

In this study, we engineered HIV-based VLPs to deliver RNP complexes of the AsCas12a nuclease and crRNA expressed from the Pol II-dependent CMV promoter. To the best of our knowledge, this is the first example of VLPs produced with a nuclease other than SpCas9. In all previous HIV- or MLV-based VLP systems, except for the ‘NanoMEDIC’ system, sgRNA was expressed from the Pol III-dependent U6 promoter [9,10,11,12]. Assembly of VLPs occurs in the cytoplasm, but U6-driven sgRNA is localized in the nucleus, thus hindering the efficient assembly and packaging of RNPs into VLPs. In this study, we adopted the packaging mechanism of ‘NanoMEDIC’ VLPs and fused AsCas12a with the FRB domain to allow its dimerization with FKBP12-Gag in the presence of a rapamycin analog, as described for SpCas9 by Gee et al. [11]. We exploited the ability of AsCas12a to excise mature crRNA from a precursor molecule and expressed crRNA as part of a Pol II-driven transcript under the control of the CMV promoter. We hypothesized that this feature of AsCas12a would assist in easily enriching crRNA in the cytoplasm, boosting RNP formation and improving their packaging into VLPs, thus increasing editing efficiency.

We found that AsCas12a-VLPs produced with CMV-driven crRNA demonstrated higher knockout rates for *Venus* and *CCR5* transgenes in comparison to VLPs with U6-driven crRNA (VLPs #2). The highest editing levels were achieved with VLPs #3 produced with the pCMV-mClover-based crRNA-expressing plasmid and were 2–3-fold higher than that for VLP #2. VLPs #4 were produced with the single plasmid coding for both AsCas12a and one copy of crRNA. The molar ratio of AsCas12a to crRNA expressed from the single plasmid is suboptimal; nevertheless, VLPs #4 outperformed VLPs #2 produced with U6-driven crRNA and were only 1.5-fold less active than VLPs #3. In Jurkat T cells, which are more difficult to transduce than 293 cells, only VLPs #3 induced reasonable editing levels and ensured 20% *CXCR4* knockout, whereas 1–2% knockout levels were observed for VLPs #2 with U6-driven crRNA and VLPs #4 produced with the single plasmid. Editing with crRNA expressed from the Pol II promoter was abolished by the H800A mutation in the RNAse domain of AsCas12a, which confirms that it was dependent on the cleavage of the crRNA-containing transcript by AsCas12a.

The editing efficiency of *CXCR4* in Jurkat T cells was further improved by up to 40% by increasing the amount of the CMV-driven crRNA-expressing plasmid and even by up to 60% and 70% by using a crRNA-expressing plasmid with an array of three or six copies of crRNA, respectively. An additional increase in Jurkat T cell editing by 1.5-fold was achieved by pseudotyping AsCas12a-VLPs by two envelop proteins, VSVG and BaEVRless, as was shown for ‘Nanoblades’ VLPs in the study by Mangeot et al. [9]. The combination of a dual-envelope coating and the plasmid with an array of six crRNAs induced the highest *CXCR4* knockout level, reaching 80%. As a result, the *CXCR4* editing efficiency was improved 80-fold in comparison to AsCas12a-VLPs with U6-driven crRNA.

In all instances, AsCas12a-VLPs produced with CMV-driven crRNA outperformed VLPs produced with U6-driven crRNA, which we explain by the increased cytoplasmic levels of CMV-driven crRNA. We did not quantify the crRNA levels in VLPs or producer cells; nevertheless, we obtained indirect evidence to support our hypothesis. First, VLPs #3, which differed from VLPs #2 only by the type of crRNA-expressing plasmid (with a CMV or U6 promoter, respectively), had the same nuclease level but a higher editing efficiency (Figure 6C,D). Second, we showed that preliminary transfection of pKS-U6-crRNA into target cells increased editing levels more substantially in samples with U6-driven crRNA than in samples with CMV-driven crRNA (Appendix A). Both results suggest that CMV-driven crRNA was incorporated into VLPs at a higher level. We achieved further improvement in *CXCR4* editing in Jurkat T cells simply by increasing the amount of the crRNA-encoding pCMV-mClover-based plasmid or by increasing the number of identical crRNAs in the transcript to up to six copies (Figure 7D,E). Taken together, these results indicate that upon VLP assembly, a fraction of AsCas12a molecules in VLPs are not bound to crRNA, not only under U6-driven crRNA expression conditions but also with CMV-driven crRNA expression from a plasmid with a single spacer, underscoring the importance of developing strategies to increase crRNA levels in the cytoplasm of VLP-producing cells. Our data suggest that using arrays with identical crRNA copies might be a viable approach. The longest array used in this study contained six identical copies of crRNA and enabled the highest editing levels. Higher numbers of crRNA copies should be tested in further studies.

Improved processing of pre-crRNA by AsCas12a can increase VLP editing efficiency by raising crRNA levels in the cytoplasm. In this study, we compared *CXCR4* knockout levels obtained for plasmids encoding crRNA arrays with canonical direct repeat sequences or sequences optimized for AsCas12a processing [34] and did not observe any differences (Figure 4A,B). Therefore, we did not apply the optimized direct repeat design for VLP production. However, direct repeat optimization may be important for other loci with a different nucleotide composition in the 3′-end of the spacer sequence, as was found by Magnusson et al. [34], and should be tested for VLP production and editing in a panel of other target loci. Other modifications of the crRNA scaffold, including base substitutions, deletions, and chemical modifications [37,38], could be tested for VLP production as well.

The ability of AsCas12a to cleave pre-crRNA is widely used for multiplexed editing [19,23]. We showed efficient simultaneous editing of *CCR5* and *CXCR4* by using a plasmid with an array of three spacers for each locus. Although we did not examine multiplexed editing with VLPs, we showed efficient editing of target loci using VLPs produced with CMV-driven arrays of up to six crRNAs. As suggested above, efficient multiplexed editing with AsCas12a VLPs might be achieved by increasing the number of crRNA copies per transcript, but this should be evaluated in further studies.

Nucleases used for genome editing are typically fused with one or several nuclear localization sequences (NLSs) that allow their enrichment in the nucleus. It reduces their cytoplasmic levels, which creates a barrier for efficient packaging into VLPs. In accordance with this logic, the insertion of three nuclear export signals (3×NES) between Gag and SpCas9 increased eVLP editing levels due to SpCas9 enrichment in the cytoplasm and improved incorporation into particles [12]. Applying the same 3×NES design to the ‘NanoMEDIC’ VLP system seems impractical since the 3xNES motif should be removed from the nuclease together with the Gag molecule by the viral protease, but ‘NanoMEDIC’ VLPs do not contain this enzyme [11].

In most VLP systems, U6-driven sgRNA is passively incorporated into particles [9,10,11,12]. Obviously, this packaging is suboptimal since it has been shown that preliminary transfection of sgRNA into target cells or inclusion of a U6-sgRNA cassette into a reporter as part of the VLP cargo increased editing levels [10,11]. The editing efficiency of VLPs with U6-driven sgRNA could be substantially improved by the insertion of the U6-sgRNA cassette in all plasmids used for Cas9-VLP production [14] or by combining Gag-SpCas9 incorporation and MS2-sgRNA packaging [13]. However, both modifications to the VLP systems increased the complexity of the design, especially the second approach, and they still relied on the passive diffusion of U6-driven sgRNA into the cytoplasm from the nucleus. In the ‘NanoMEDIC’ system, SpCas9 sgRNA was expressed as part of the Pol II transcript under the control of the LTR promoter. The transcript was expressed in the presence of the HIV protein Tat, transported into the cytoplasm, and packaged into particles due to the interaction between the Ψ packaging signal and Gag. This was followed by ribozyme-mediated sgRNA excision [11]. However, the efficiency of sgRNA processing by ribozymes and its cytoplasm localization have not been investigated. The contribution of this multicomponent transcript in editing levels was abolished by the removal of the Ψ packaging signal, which suggests that free sgRNA content in the cytoplasm produced by ribozymes was insufficient for packaging [11]. Another strategy to localize SpCas9 sgRNA into the cytoplasm is based on the use of Csy4 endonuclease, but it has not yet been explored for VLP design [16]. Similar to the ‘NanoMEDIC’ system, which relies on the presence of Tat, this strategy requires the overexpression of an additional protein, Csy4, which is undesirable for VLP production [11,16].

Here, we showed that in contrast to the CMV-driven transcript with AsCas12a crRNA, the CMV-driven transcript coding for SpCas9 sgRNA flanked by ribozymes could not efficiently induce *CXCR4* editing, neither upon plasmid electroporation nor VLP delivery (Figure 3G, Appendix A). In the AsCas12a-VLPs generated in our study, crRNA is directed into the cytoplasm due to its inclusion in the Pol II transcript, whereas its processing by AsCas12a and subsequent association with the nuclease enables its packaging into VLPs. Our data suggest that this mechanism is more efficient than the mechanism of sgRNA processing by ribozymes. Additionally, AsCas12a crRNA expression does not require the HIV protein Tat, thus reducing the complexity of the VLP design and eliminating potential safety risks associated with Tat [11].

In our study, AsCas12a-VLPs outperformed SpCas9-VLPs produced not only with CMV-driven gRNA (discussed above) but also with U6-driven gRNA. The similar content of both nucleases in particles revealed by Western blot analysis allows us to assume that this was not caused by a smaller size of AsCas12a in comparison to SpCas9, boosting its packaging into VLPs. Moreover, when the nucleases were delivered into cells in the form of plasmids, SpCas9 demonstrated similar or superior editing of these loci in comparison to AsCas12a. We suppose that in the case of U6-driven gRNA, the increased activity of AsCas12a-VLPs could be attributed to better packaging of crRNA in comparison to sgRNA. Presumably, it depends on the crRNA transcription level, which may be higher than that of longer sgRNA molecules and may affect the resultant level of gRNA in the cell [39]. Additionally, the binding affinity of crRNA to AsCas12a may be higher, as compared to the sgRNA/SpCas9 interaction, which influences its packaging [40,41]. To test this hypothesis, the levels of crRNA and sgRNA should be quantified in producer cells and in VLPs for both nucleases.

One of the important questions regarding RNP incorporation into VLPs is the specificity of packaging. Unexpectedly, we found that AsCas12a lacking the FRB domain and expressed together with U6-driven crRNA was incorporated into VLPs, although at a lower level, as compared to FRB-AsCas12a. In 293 cells, these control VLPs induced editing levels that were only 1.7-fold lower than those for VLPs with FRB-AsCas12a (Figure 6). However, in Jurkat T cells, the control VLPs showed minimal editing, which can be explained by the less efficient transduction of this cell line. These data, together with the Western blot results, indicate that a substantial fraction of RNP molecules is passively incorporated into VLPs and induces reasonable editing levels in cell lines that are easy to transduce. Not only nucleases and RNPs are subject to nonspecific packaging, but also various cellular proteins and RNA molecules, such as microRNAs and other short RNAs [9]. This may lead to undesirable effects on target cells, which is especially important to consider when developing therapeutic approaches. The protein and RNA content of therapeutic VLPs should be carefully analyzed using high-throughput techniques such as NGS and mass spectrometry.

The target genes *CCR5* and *CXCR4* used in our study are relevant for HIV gene therapy. The knockout of these receptors confers cell resistance to infection [24,42]. Many research groups, including ours, have developed strategies to disrupt *CCR5* and/or *CXCR4*, but to the best of our knowledge, VLP delivery has not been used for this purpose [27,43,44,45]. Given its lower toxicity compared to electroporation, VLPs can be advantageous for targeting primary cells. Better protection from HIV is achieved by the simultaneous knockout of coreceptors and knock-in of an HIV restriction factor or peptide fusion inhibitor [24]. This should be tested in further experiments by combining AsCas12a-VLPs with donor DNA electroporation or donor AAV6 transduction. Thus, AsCas12a-VLPs represent a promising modality for the development of therapeutic approaches to eradicate HIV or treat other diseases.

We acknowledge that our study has several limitations. First, in future studies, the efficiency of AsCas12a VLPs should be tested in a broader panel of human primary cells and cell lines. Second, the editing targets included two reporter genes and one endogenous locus; therefore, AsCas12a-VLPs must be tested in a wider panel of endogenous loci. Third, we did not analyze the off-target editing of AsCas12a-VLPs and their toxicity, which also must be compared to that of the original ‘NanoMEDIC’ VLPs with SpCas9. Increased AsCas12a specificity over SpCas9 reported in a genome-wide study [22] should be confirmed for VLPs. Fourth, the level of crRNA in producer cells and in VLPs was not directly measured. To address this issue, it is required to develop a sensitive and reliable assay for the quantification of crRNA by qPCR or ddPCR. We would like to emphasize that our results indirectly indicate that CMV-driven crRNA is more efficiently packaged into VLPs compared to U6-driven crRNA or sgRNA. Fifth, additional parameters of VLPs should be analyzed, such as particle size, RNP content per particle, the content of passively packaged cellular proteins and RNA molecules, and the presence of residual plasmid DNA since it may be toxic to target cells. Sixth, we used a different protocol for VLP production compared to ‘NanoMEDIC’ VLPs [11]. Therefore, for the direct comparison of AsCas12a-VLPs with original ‘NanoMEDIC’ VLPs, particles must be produced by the same protocol. Additionally, we produced VLPs only on a small scale, and for therapeutic applications, the protocol requires further optimization and scaling up. Another limitation, which is inherent to the ‘NanoMEDIC’ packaging system, is the requirement for a costly dimerizing molecule, AP21967, which has not yet been approved for clinical use. Although the authors of the ‘NanoMEDIC’ VLP system demonstrated its scalability and compatibility with xeno-free conditions, it can be a disadvantage for manufacturing VLPs on a large scale. All these issues should be addressed in further research.

Taken together, we showed the possibility of AsCas12a incorporation into VLPs with concomitant pre-crRNA cleavage and the formation of active RNPs. To this end, we used the pre-crRNA processing activity of AsCas12a to localize crRNA in the cytoplasm where VLP assembly takes place, which substantially increased VLP editing efficiency. Our results suggest that the ability of AsCas12a to process pre-crRNA could be beneficial for VLP production, thus making AsCas12a a preferred nuclease for VLP packaging compared to SpCas9. Based on this idea, one might predict that the development of an efficient method to direct SpCas9 sgRNA into the cytoplasm will also boost the editing efficiency of SpCas9-VLPs. We believe that our results open new avenues for the development of other VLP systems such as eVLP [12] or Cas-VLP [10] with AsCas12a and CMV-driven crRNA. Moreover, other nucleases of the Cas12a family, such as LbCas12a [46] or MbCas12a [47], as well as the highly active AsCas12a Ultra variant with improved specificity [48], may be evaluated as VLP cargos. AsCas12a-based prime [49] and base [50] editors may also be applied. Additionally, due to easy multiplexing, AsCas12a-VLPs represent an attractive tool for the simultaneous editing of multiple loci. We suggest that AsCas12a-VLPs will broaden the genome editing toolkit and will allow for efficient editing in mammalian cells.

## 4. Materials and Methods

### 4.1. Cell Culture

Human embryonic kidney 293T cells were obtained through the NIH AIDS Research and Reference Reagent Program. 293T/CD4/CCR5 cells, clone #19, were generated through sequential stable lentiviral transduction of 293T cells, as described in [24], followed by cell cloning with fluorescence-activated cell sorting. CEM/CCR5 cells were derived from the CCRF-CEM cell line (Cellosaurus CVCL_0207, purchased from the ATCC, Manassas, VA, USA) [8], and CEM/CCR5 clone #8 cells were produced from CEM/CCR5 cultures by cell cloning with fluorescence-activated cell sorting. Jurkat T cells (Cellosaurus CVCL_0065) were purchased from the ATCC (Manassas, VA, USA). All cell lines were cultured in DMEM/F12 (PanEco, Moscow, Russia) supplemented with 10% fetal bovine serum (#SV30160.03, HyClone, Cytiva, Marlborough, MA, USA), 4 mM L-glutamine (PanEco, Moscow, Russia), and 10 µg/mL gentamycin (PanEco, Moscow, Russia) at 37 °C in a 5% CO_2_ humidified atmosphere.

### 4.2. Generation of 293-Venus Reporter Cell Line

293-Venus clone #8 cells were derived from 293 cells by CRISPR/Cas9-mediated insertion of the Traffic Light Reporter 5 [25] into the *AAVS1* locus, with subsequent correction of the *Venus* sequence via knock-in.

First, we generated the HEK293-TLR5 reporter cell line via knock-in of the pAAVS-TLR plasmid (#64215, Addgene, [25]) into the AAVS1 locus of HEK293 cells (European Collection of Cell Cultures, #85120602). To this end, 0.5 × 10^6^ HEK293 cells were seeded in a 35 mm plate and, 24 h later, transfected with the donor plasmid pAAVS-TLR and the plasmid pX458-sgAAVS2 encoding sgAAVS1-2 ([25]) in 1:1 ratio using the EcoTransfect transfection reagent (#ET11000, OZ Bioscience SAS, Marseille, France) according to the manufacturer’s protocol. Three days after transfection, cells were transferred to a 100 mm plate and pools of stably transfected cells were selected with 0.5 µg/mL of puromycin for 15 days. Next, single-cell clones were obtained by limiting dilution. A total number of 36 clones were collected and analyzed for correct knock-in of the TLR5 reporter construct by PCR. Correct knock-in was confirmed in 12 clones. In addition, these clones were shown to be negative for random integration of pAAVS-TLR and pX458-sgAAVS2 plasmids by PCR with backbone-specific primers and qPCR for SpCas9 cDNA, respectively.

Next, to generate the 293-Venus reporter cell line, we selected HEK293-TLR5 clone #31, which was characterized in terms of genome editing. To this end, 0.1 × 10^6^ HEK293-TLR5 clone #31 cells were seeded to a well of a 24-well plate and transfected with pX330-sgR26-3 (expressing sgRNA to the mouse *Rosa26* locus) and pTLR-donor-∆ATG∆polyA plasmids (pTLR-repair plasmid lacking the ATG codon for Venus and SV40 polyA signal) in a 3:5 ratio using the Lipofectamine 2000 transfection reagent (#11668019, Invitrogen, Thermo Fisher Scientific, Waltham, MA, USA) according to the manufacturer’s protocol. Three days after transfection, Venus+ cells were sorted as a pool using a Sony MA-900 cell sorter (Sony Biotechnology Inc., San Jose, CA, USA). Sorted cells were next expanded in the presence of 0.5 µg/mL puromycin and single-cell clones were generated by limiting dilution. A total number of 16 clones were obtained and clone #8 was selected as the 293-Venus reporter cell line on the basis of robust growth and efficient *Venus* editing (Appendix A.

### 4.3. Plasmid Construction

A list of guide RNA target sequences is provided in Appendix A. Oligonucleotides used for cloning are listed in Appendix A, and the plasmids used and obtained in this study are shown in Appendix A. All restriction endonucleases and other enzymes used for cloning are from SibEnzyme (SibEnzyme, Novosibirsk, Russia) if not stated otherwise. Nucleotide sequences of all plasmid constructs were verified by Sanger sequencing, which was performed by Evrogen (Evrogen, Moscow, Russia). All oligonucleotides were synthesized by Evrogen. Plasmid DNA was isolated using the Plasmid Midiprep 2.0 kit (#BC124, Evrogen). DNA fragments from agarose gel were purified using the Cleanup S-Cap kit (#BC041L, Evrogen). Additional cloning details are available upon reasonable request.

#### 4.3.1. Plasmids for the Generation of the 293-Venus Reporter Cell Line

pTLR-donor-∆ATG∆polyA was generated on the basis of a pTLR-repair donor plasmid (Addgene #64322, [25]). First, the polyA tail was deleted by SacI digestion and self-ligation of the pTLR-repair donor. Next, a Venus cDNA fragment was amplified with primers 5′-VenRI and 3′-VenSp, and a HindIII/PstI fragment was cloned into pTLR-repair ∆pA.

#### 4.3.2. AsCas12a Vector Cloning

The sequence of AsCas12a was cloned from pET-28b-T7-henAsCas12a-HF1(E174R/N282A/S542R/K548R)-NLS(nucleoplasmin)-6xHis (#114073, Addgene, [33]) to the pCMVpA vector for mammalian expression through several subcloning steps. The resultant plasmid was named pCMV-AsCas12a-2xNLS. To add a 3xHA tag at the C-terminus of AsCas12a, four annealed oligonucleotides, 5′-NotI-3xHA-1, 5′-NotI-3xHA-2, 3′-XmaI-3xHA-1, and 3′-XmaI-3xHA-2, were cloned into the NotI/XmaI-treated plasmid pCMV-AsCas12a-2xNLS. The obtained plasmid, pCMV-AsCas12a-2xNLS-3xHA, was later referred to as pCMV-AsCas12a for simplicity.

To clone the FRB domain at the N-terminus of AsCas12a, the FRB coding fragment was amplified from pHLS-FRB-SpCas9 (#138477, Addgene, [11]) with the primers 5′-XbaI-FRB and 3′-FRB-NheI-Cas12. The N-AsCas12a coding fragment was amplified from pCMV-AsCas12a with the primers 5′-Cas12 and 3′-Cas12a-BlpI. Both fragments were then combined by overlap extension PCR, and the product was cloned in the pJET1.2 vector (Thermo Fisher Scientific, Waltham, MA, USA). The XbaI/BlpI fragment of pJET1.2 was cloned into the corresponding sites of pCMV-AsCas12a to obtain pCMV-FRB-AsCas12a.

To clone the FRB domain at the C-terminus of AsCas12a, the FRB coding fragment was amplified from a HindIII/SalI fragment of pHLS-FRB-SpCas9 (#138477, Addgene, [11]) with the primers 5′-NotI-NLS-NheI-FRB and 3′-FRB-BamHI, and cloned in pJET1.2. Next, the NotI/BamHI fragment was cloned from pJET1.2 into the corresponding sites of pCMV-AsCas12a to obtain pCMV-AsCas12a-FRB.

To generate the AsCas12a H800A RNase-deficient mutant, a StuI/StuI fragment was cloned from pCMV-AsCas12a into a StuI-digested pCR-Blunt vector (Thermo Fisher Scientific, Waltham, MA, USA). Next, the H800A mutation (cac → gcc) was introduced with 5′-H800A and 3′-H800A primers, using the QuikChange site-directed mutagenesis protocol with a PrimeSTAR HS PCR premix (#R040, Takara Bio Inc., Kusatsu, Shiga, Japan). Next, the StuI/StuI fragment with the H800A mutation was cloned from pCR-Blunt into the pCMV-AsCas12a plasmid.

#### 4.3.3. Guide RNA Cloning for U6-Driven Expression

SpCas9 sgRNA cloning was performed according to the protocol described by Ran et al. [51] using a pKS-U6-sgRNA-BB vector [24], pX330 (#42230, Addgene, [52]), and pX458 (#48138, Addgene, [51]). For AsCas12a crRNA cloning, the BbsI/XbaI fragment in the pKS-U6-sgRNA-BB vector was replaced with a pair of annealed oligonucleotides, a 5′-crRNA-scaffold and a 3′-crRNA-scaffold, to produce pKS-U6-crRNA-BB and allow BbsI-mediated cloning of crRNA-encoding oligonucleotides.

#### 4.3.4. Guide RNA Cloning for CMV-Driven Expression

To clone the plasmid pCMV-mClover-trpl-Rib-sg1X4-Rib, where the SpCas9 sgRNA spacer for *CXCR4* is flanked by ribozymes, the sequences of HH and HDV ribozymes were used according to the plasmid construct PL-5LTR-RGR(DMD#1)-AmCyan-A (#138482, Addgene), as described by Gee at al. [11]. First, annealed oligonucleotides 5′-MCS-NM and 3′-MCS-NM were cloned into BsrGI/PspXI-treated pUCHR-mClover-MT-C34 (#177153, Addgene, [24]) to produce an intermediate plasmid, pUCHR-mClover-MCS, with a multiple cloning site. Next, annealed oligonucleotides 5′-XmaI-Trpl and 3′-MluI-Trpl were cloned into XmaI/MluI-treated pUCHR-mClover-MCS to introduce the Triplex sequence according to Campa et al. [23]. At the next step, the HH-sg1X4-HDV fragment was amplified from the pKS-U6-sgRNA-X4ex2 plasmid [24] with 5′-HH-X4 and 3′-X4-HDV primers, treated with MluI/PspXI, and cloned into the corresponding sites of pUCHR-mClover-trpl-MCS. Finally, the SnaBI/PspXI fragment from pUCHR-mClover-trpl-HH-sg1X4-HDV was cloned into the corresponding sites of pCMV-AsCas12a.

To obtain pCMV-mClover-trpl-DR-cr1X4-DR, where the AsCas12a crRNA spacer #3 for *CXCR4* is flanked by direct repeats, the annealed oligonucleotides 5′-crRNA-X4-crRNA and 3′-crRNA-X4-crRNA were cloned into MluI/PspXI-treated pCMV-mClover-trpl-Rib-sg1X4-Rib. To obtain pCMV-mClover-trpl-DR-cr1R5-DR with the AsCas12a crRNA spacer #2 targeting *CCR5*, the annealed oligonucleotides 5′-crRNA-R5-crRNA and 3′-crRNA-R5-crRNA were cloned into MluI/PspXI-treated pCMV-mClover-trpl-DR-cr1X4-DR. To obtain pCMV-eBFP2-trpl-DR-cr1Venus-DR with the AsCas12a crRNA spacer targeting *Venus*, the annealed oligonucleotides 5′-crRNA-Venus-crRNA and 3′-crRNA-Venus-crRNA were cloned into MluI/PspXI-treated pCMV-mClover-trpl-DR-cr1X4-DR. The sequence of mClover in the resultant vector was then replaced with that of eBFP2. To this end, the eBFP2 coding fragment was amplified from pU6-(BbsI)-CBh-Cas9-T2A-BFP (#64323, Addgene, [25]) with the primers 5′-PspXI-XbaI-eBFP2 and 3′-eBFP2-ClaI, and cloned into pJET1.2. The PspXI/ClaI fragment of pJET1.2 was then cloned into SalI/ClaI-treated pCMV-mClover-trpl-DR-cr1Venus-DR to produce pCMV-eBFP2-trpl-DR-cr1Venus-DR.

pCMV-mClover-DR-cr1X4-DR and pCMV-mClover-Rib-sg1X4-Rib without Triplex were obtained from the corresponding Triplex-containing plasmids by sequential treatment with ClaI/MluI and a DNA Polymerase I Klenow Fragment, followed by ligation of blunt ends.

To clone plasmids with 6 spacers, the fragment containing 6 crRNAs for *CXCR4*, 6 crRNAs for *CCR5*, and an array of alternating 3 *CCR5* and 3 *CXCR4* crRNAs were synthesized as a part of the pUC57Kan-trpl-6X4-6R5-3X43R5 vector by Synbio Technologies (Monmouth Junction, NJ, USA). The SmaI/XhoI fragment of pUC57Kan-trpl-6X4-6R5-3X43R5 containing Triplex-cr6X4 was cloned into the corresponding sites of pCMV-mClover-trpl-DR-cr1X4-DR to produce pCMV-mClover-trpl-cr6X4 (DR is omitted for simplicity). The BamHI/SalI fragment from pUC57Kan-trpl-6X4-6R5-3X43R5 containing 6R5 was first cloned into BamHI/XhoI-treated pBluescript KS(+) (Stratagene, Agilent Technologies, Inc., Santa Clara, CA, USA) to generate the pBl-6R5 plasmid, followed by AflII/XhoI fragment cloning from pBl-6R5 into the corresponding sites of pCMV-mClover-trpl-cr6X4 to produce pCMV-mClover-trpl-cr6R5. Finally, the BglII/EcoRI fragment from pUC57Kan-trpl-6X4-6R5-3X43R5 containing 3X43R5 was first cloned into BamHI/EcoRI-treated pBluescript KS(+) to generate the pBl-3X43R5 plasmid, followed by AflII/XhoI fragment cloning into the corresponding sites of pCMV-mClover-trpl-cr6X4 to produce pCMV-mClover-trpl-cr3X4_cr3R5.

To generate a plasmid with 3 spacers for *CXCR4* (3X4), the XmaI/PspXI fragment from the plasmid pCMV-AsCas12a-trpl-DR-cr3X4-DR (described in the next section) was cloned into the corresponding sites of pCMV-mClover-trpl-DR-cr1X4-DR to produce pCMV-mClover-trpl-cr3X4. To generate a plasmid with 3 spacers for *CCR5* (3R5), 3 copies of the crRNA coding sequence flanked by DR were amplified on a template of the pBl-6R5 plasmid with the primers 5′-3R5 and 3′-3R5. PCR products were resolved in agarose gel and a gel fragment of the size that corresponds to 3R5 (145 nt) was excised. DNA was purified, reamplified in PCR with the same primers, cloned into a pCR-Blunt vector, and sequenced. Next, the AflII-Xho fragment was cloned from pCR-Blunt into the corresponding sites of pCMV-mClover-trpl-cr6X4 to produce pCMV-mClover-trpl-cr3R5.

#### 4.3.5. Cloning of Single Constructs for AsCas12a and crRNA Expression

To produce the single plasmid pCMV-AsCas12a-trpl-DR-cr1X4-DR, the XmaI/PspXI fragment from pCMV-mClover-trpl-DR-cr1X4-DR was cloned into the corresponding sites of pCMV-AsCas12a.

To obtain single plasmids with the 50 nt insert between Triplex and the first direct repeat of crRNA, the oligonucleotides 5′-insert-MluI and 3′-insert were cloned into MluI-treated pCMV-AsCas12a-trpl-DR-cr1X4-DR. The sequence for this insert corresponded to the non-coding fragment between Triplex and DR in a single AsCas12a plasmid described by Campa et al. [23]. For the cloning of the 150 nt insert #1, the non-coding fragment from the pcDNA3.1-hygro backbone (Invitrogen, USA) was amplified with the primers 5′-pcDNA3.1-hygro-insert and 3′-pcDNA3.1-hygro-insert. For the cloning of the 150 nt insert #2, the EGFP fragment from pEGFP-N1 (Clontech, Takara Bio Inc., Kusatsu, Shiga, Japan) was amplified with the primers 5′-EGFP-insert and 3′-EGFP-insert. PCR products were treated with MluI and cloned into MluI-treated pCMV-AsCas12a-trpl-DR-cr1X4-DR. For the EGFP-derived insert, a plasmid with the orientation opposite to the EGFP coding sequence was chosen.

Generation of plasmids with two and three copies of the spacer was performed using the protocol described by Scior et al. for the cloning of repeated sequences [53]. First, to use the BbsI site for cloning, the BbsI site present in the sequence of the CMV promotor in pCMV-AsCas12a-trpl-DR-cr1X4-DR was mutated. To this end, the fragment of the CMV promoter containing the BbsI site was amplified with the primers 5′-SnaBI and 3′-BbsI-mut, and cloned into pJET1.2. The SnaBI/BbsI fragment from pJET1.2 was then cloned into the corresponding sites of pCMV-AsCas12a-trpl-DR-cr1X4-DR to produce pCMV(BbsI-mut)-AsCas12a-trpl-DR-cr1X4-DR. This plasmid was used to generate plasmids with 2 and 3 spacers and plasmids with modified direct repeats.

For cloning of single plasmids with 2 and 3 spacers, first, the annealed oligonucleotides 5′-Mlu-DR-X4-DR-BbsI-Acc65I and 3′-Mlu-DR-X4-DR-BbsI-Acc65I were cloned into MluI/Acc65I-treated pCMV(BbsI-mut)-AsCas12a-trpl-DR-cr1X4-DR to produce the intermediate plasmid pCMV(BbsI-mut)-AsCas12a-trpl-DR-cr1X4-DR-BbsI. Next, the annealed oligonucleotides 5′-DR-X4-DR-BbsI-Acc65I and 3′-DR-X4-DR-BbsI-Acc65I were cloned into BbsI/Acc65I-treated pCMV(BbsI-mut)-AsCas12a-trpl-DR-cr1X4-DR-BbsI to produce the intermediate plasmid pCMV(BbsI-mut)-AsCas12a-trpl-DR-cr1X4-DR-cr1X4-DR-BbsI. Alternatively, the annealed oligonucleotides 5′-DR-X4-DR-Acc65I and 3′-DR-X4-DR-Acc65I were cloned into BbsI/Acc65I-treated pCMV(BbsI-mut)-AsCas12a-trpl-DR-cr1X4-DR-BbsI to produce the plasmid pCMV(BbsI-mut)-AsCas12a-trpl-DR-cr1X4-DR-cr1X4-DR with 2 copies of the spacer. Finally, the annealed oligonucleotides 5′-DR-X4-DR-Acc65I and 3′-DR-X4-DR-Acc65I were cloned into BbsI/Acc65I-treated pCMV(BbsI-mut)-AsCas12a-trpl-DR-cr1X4-DR-cr1X4-DR-BbsI to produce the plasmid pCMV(BbsI-mut)-AsCas12a-trpl-DR-cr1X4-DR-cr1X4-DR-cr1X4-DR with 3 copies of the spacer. For simplicity, plasmids with 2 and 3 spacers were then referred to as pCMV-AsCas12a-trpl-DR-cr2X4-DR and pCMV-AsCas12a-trpl-DR-cr3X4-DR. Single plasmids with arrays containing an optimized sequence of direct repeats were generated similarly to plasmids with the original direct repeat, using corresponding oligonucleotides with additional 4T nucleotides at the 5′-end of direct repeats [34] (oligonucleotide sequences are listed in Appendix A). To restore the BbsI site in the resultant single plasmids for correct comparison with other plasmids, the SnaBI/XbaI fragment of the CMV promoter from pCMV-AsCas12a was cloned into the corresponding sites of single plasmids with the mutated BbsI site.

To add the 3xHA sequence at the C-terminus of AsCas12a in the single plasmid coding for crRNA against *CXCR4*, the NotI/XmaI fragment from pCMV-AsCas12a coding for SV40 NLS and 3xHA was cloned into the corresponding sites of pCMV-AsCas12a-trpl-DR-cr1X4-DR.

To add the FRB domain to the N-terminus of AsCas12a in the single plasmid coding for crRNA against *CXCR4*, the XbaI/BlpI fragment from pCMV-FRB-AsCas12a coding for FRB was cloned into the corresponding sites of pCMV-AsCas12a-trpl-DR-cr1X4-DR. For cloning of single plasmids with the FRB domain targeting *CCR5* and *Venus*, the XmaI/PspXI fragments from pCMV-mClover-trpl-DR-cr1R5-DR and pCMV-eBFP2-trpl-DR-cr1Venus-DR, respectively, were cloned into the corresponding sites of pCMV-FRB-AsCas12a-trpl-DR-cr1X4-DR.

### 4.4. Electroporation

For plasmid electroporation, 1.5 × 10^6^ CEM/CCR5 or Jurkat T cells were electroporated using a Neon electroporation system with 100 µL tips (Invitrogen, Thermo Fisher Scientific, Waltham, MA, USA), using the following settings: 1230 V, 40 ms, and 1 pulse for CEM/CCR5 cells, and 1350 V, 10 ms, and 3 pulses for Jurkat T cells. If not otherwise stated, 0.6 pmol (3.5 µg) of pcDNA3.3-hSpCas9 (#41815, Addgene, [54]) with 0.48 pmol (1 µg) of pKS-U6-sgRNA or 0.6 pmol (2.9 µg) of pCMV-AsCas12a with 0.48 pmol (1 µg) of pKS-U6-crRNA were used. Plasmid quantities used for CMV-driven crRNA expression and single constructs are indicated in the corresponding sections. Knockout levels were assessed by flow cytometry on day 5 after electroporation.

### 4.5. RNP Production

RNP complexes were assembled using the SpCas9 and AsCas12a produced and purified as described previously [24]. Synthesis of gRNA was carried out via in vitro transcription with a HiScribe T7 High Yield RNA synthesis kit (#E2040S, New England Biolabs, Ipswich, MA, USA) as described by [24] (spacer sequence #3 was used for AsCas12a crRNA). For electroporation, 200 pmol of nuclease was combined with 200 pmol of gRNA in buffer R (Invitrogen, Thermo Fisher Scientific, Waltham, MA, USA), and the mixture was incubated at room temperature for 20 min followed by electroporation.

### 4.6. Transfection

To assess the level of AsCas12a and its variants expressed from different constructs, 293T cells were seeded in 24-well plates at a density of 1 × 10^5^ per well in 0.5 mL of full medium. The next day, cells were transfected with 0.25 µg of DNA and 0.5 µL of the transfection reagent GenJect39 (Molecta, Moscow, Russia) mixed in OptiMEM medium (#31985070, Thermo Fisher Scientific, Waltham, MA, USA) according to the manufacturer’s instructions. After 4 h, the medium was replaced with fresh pre-warmed medium.

To evaluate the editing efficiency of plasmid constructs, 293-Venus clone #8 and 293T/CD4/CCR5 clone #19 were seeded in 24-well plates at a density of 3 × 10^5^ or 1 × 10^5^ per well, respectively, in 0.5 mL of full medium. The next day, cells were transfected with 0.06 pmol (~0.3 µg) of the AsCas12a plasmid and 0.048 pmol (~0.1 µg) of the crRNA plasmid with 0.5 µL of GenJect39 (Molecta, Moscow, Russia) according to the manufacturer’s instructions. Single plasmids were used at doses of 0.09 pmol (~0.5 µg) and 0.12 pmol (~0.6 µg). After 4 h, the medium was replaced with fresh pre-warmed medium.

For preliminary transfection of target cells, 293-Venus clone #8 and 293T/CD4/CCR5 clone #19 were seeded in 12-well plates at a density of 1 × 10^6^ or 3 × 10^5^ per well, respectively, in 1 mL of full medium. The next day, cells were transfected with 0.5 µg of pKS-U6-crRNA/sgRNA. pKS-U6-crRNA/sgRNA with an irrelevant spacer sequence was used as a control.

### 4.7. VLP Production

293T cells were seeded in 9 cm dishes at a density of 3.2 × 10^6^ per plate in 10 mL of full medium. The next day, cells were transfected with GenJect39 (Molecta, Moscow, Russia) in a 1:2 (DNA, µg to GenJect, µL) ratio according to the manufacturer’s instructions using OptiMEM medium (#31985070, Thermo Fisher Scientific, Waltham, MA, USA).

For comparison of VLPs with SpCas9 and AsCas12a, cells were transfected with 0.28 µg pCMV-VSVG (#8454, Addgene [55]), 1.24 µg of pHLS-FKBP12-Gag (#138476, Addgene, [11]), 1.24 µg of pKS-U6-sgRNA or pKS-U6-crRNA, and 1.24 µg of pCMV-FRB-AsCas12a or pHLS-FRB-SpCas9 (#138477, Addgene, [11]).

For comparison of four types of VLPs with AsCas12a, cells were transfected with 0.28 µg pCMV-VSVG with 1.24 µg of pHLS-FKBP12-Gag and one of the following plasmid combinations: (1) 1.24 µg of pKS-U6-crRNA and 1.24 µg of pCMV-AsCas12a, (2) 1.24 µg of pKS-U6-crRNA and 1.24 µg of pCMV-FRB-AsCas12a, (3) 1.66 µg of pCMV-mClover/eBFP2-trpl-DR-crRNA-DR and 1.24 µg of pCMV-FRB-AsCas12a, or (4) 2.54 µg of pCMV-FRB-AsCas12a-trpl-DR-crRNA-DR. The doses of 1.24 µg of pKS-U6-crRNA and 1.66 µg of pCMV-mClover/eBFP2-trpl-DR-crRNA-DR both represented 0.6 pmol. The dose of 2.54 µg of the pCMV-FRB-AsCas12a-trpl-DR-crRNA-DR plasmid was equal to 0.5 pmol and was two times higher than 1.24 µg (0.25 pmol) of pCMV-AsCas12a. This dose was chosen according to the data on the editing efficiency of single plasmids.

For VLP pseudotyping with BaEVRless, 0.14 µg of pCMV-VSVG and 0.14 µg of pCMV-BaEVRless were used [56].

After 4 h, the medium was replaced with fresh pre-warmed medium supplemented with 300 nM AP21967 (#635055, Takara Bio Inc., Kusatsu, Shiga, Japan). The AP21967 concentration was similar to that reported by Gee et al. [11]. Supernatants with VLPs (~9 mL) were collected 48 h after transfection, centrifuged at 350 *g* for 5 min, filtered through a 0.45 μm PVDF syringe filter (#FPV403025, Guangzhou Jet Bio-Filtration Co., Ltd., Guangzhou, China), aliquoted in 1.5 mL tubes, and centrifuged at 21,000× *g* for 2.5 h (4 °C) in a microcentrifuge. Pellets with VLPs were resuspended in OptiMEM (#31985070, Thermo Fisher Scientific, Waltham, MA, USA), combined into one tube, and adjusted to 180 μL with OptiMEM (approximate concentration by 50-fold). An aliquot of each VLP sample was taken for Western blot analysis, and the rest of the sample was treated with 1.8 μL of DNAse I (#M0303S, New England Biolabs, Ipswich, MA, USA) with the addition of the corresponding 10× DNAse I buffer by incubation at 37 °C for 10 min. Next, VLP samples were aliquoted and stored at −70 °C.

### 4.8. VLP Transduction

For 293-Venus clone #8, 293T/CD4/CCR5 clone #19 cells were detached by trypsin, centrifuged, and resuspended in OptiMEM (#31985070, Thermo Fisher Scientific, Waltham, MA, USA). CEM/CCR5 and Jurkat T cells were centrifuged and resuspended in OptiMEM. Next, 4.5 × 10^4^ cells in 40 µL were added to 50 µL of VLPs, mixed by pipetting, and cultured in a 96-well plate. After 4 h, full pre-warmed medium was added to the wells. Cells were subcultured if necessary and analyzed on day 6 by flow cytometry.

### 4.9. Flow Cytometry

Immunofluorescence staining was performed by incubating cells with a mouse monoclonal antibody against CXCR4 (#sc-12764, clone 12G5, Santa Cruz Biotechnology, Dallas, TX, USA) or CCR5 (#555991, BD Biosciences, Franklin Lakes, NJ, USA) in phosphate-buffered saline (PBS) at 4 °C for 30 min. Cells were washed twice with PBS and incubated with goat anti-mouse IgG antibodies conjugated with Alexa488 or Alexa546 (#A11001 and #A11003, respectively, Thermo Fisher Scientific, Waltham, MA, USA) at 4 °C for 30 min. Then, the cells were washed twice with PBS and examined using a CytoFLEX S flow cytometer (Beckman-Coulter, Brea, CA, USA). Cells treated only with secondary antibodies were used as a negative control. 293-Venus clone #8 cells were detached by trypsin, washed with PBS, and analyzed without staining. Data were acquired and analyzed by CytExpert 2.0 software (Beckman-Coulter, Brea, CA, USA).

### 4.10. Western Blotting

To assess the level of AsCas12a and its variants expressed from different constructs, as well as the content of SpCas9 or AsCas12a in 293T producer cells, 293T cells were lysed 48 h after transfection in buffer (50 mM Tris-HCl, pH 8.0, 150 mM NaCl, 5 mM EDTA, 1% (*w*/*v*) Triton Х-100, and 1 mM phenylmethylsulfonyl fluoride) and incubated at 4 °C for 15 min. Lysates were centrifuged at 12,000× *g* at 4 °C for 10 min, supernatants were mixed with a 4× SDS-PAGE sample buffer (250 mM Tris-HCl, pH 6.8, 40% glycerol, 8% SDS, 4% 2-mercaptoethanol, and 0.2% Bromphenol Blue), and incubated at 80 °C for 5 min. To evaluate the level of nuclease in VLPs, concentrated particles were lysed directly in 4× SDS-PAGE sample buffer and incubated at 80 °C for 5 min.

SDS-PAGE in 10% gel was carried in the Laemmli system followed by semi-dry protein transfer onto a PVDF membrane using a Trans-Blot Turbo system (Bio-Rad, Hercules, CA, USA). The membrane was blocked with 5% dry skimmed milk in PBS supplemented with 0.1% Tween-20. AsCas12a with the 3× HA epitope was detected using an anti-HA rabbit monoclonal antibody (clone C29F4, #3724, Cell Signaling Technology, Danvers, MA, USA); α-tubulin, which served as a loading control for cell lysates, was detected using a mouse monoclonal antibody (clone 12G10, Sorbent, Moscow, Russia); and p17 (Gag), which served as a loading control for VLP samples, was detected using a mouse monoclonal antibody (p17 HIV Type 1, clone 32/5.8.42, #0801005, Zeptometrix, Buffalo, NY, USA). Horseradish peroxidase-conjugated goat polyclonal antibodies against rabbit IgG (#7074, Cell Signaling Technology, Danvers, MA, USA) or mouse IgG (#7076, Cell Signaling Technology, Danvers, MA, USA) were used as secondary antibodies. The chemiluminescence signal was detected using a ChemiDoc MP imaging system (Bio-Rad, Hercules, CA, USA) and the Immobilon reagent (Millipore, Merck, Burlington, MA, USA).

### 4.11. Luciferase Assay

Pseudoviruses (PVs) with the intron-regulated reporter vector inLuc were produced as described in [24,57]. Briefly, 3 × 10^6^ 293T cells were plated in a 10 cm dish in 10 mL of growth medium. The next day, the cells were transfected with 4 µg of pCMV-dR8-2, 6 µg of pUCHR-inLuc-mR, and 1 µg of pCMV-VSVG using GenJect-39 (Molecta, Moscow, Russia) according to the manufacturer’s instructions. Then, 48 h post-transfection, PVs were cleared through 0.45 μm filters, concentrated by centrifugation at 21,000× *g*, 4 °C, 2.5 h, aliquoted, and stored at −70 °C. 293T/CD4/CCR5 clone #19 cells were plated at 5 × 10^4^ cells per well in a 96-well plate. The next day, PVs were serially two-fold diluted and added to wells for transduction in a monolayer. In parallel, 293T/CD4/CCR5 clone #19 cells were detached with trypsin and transduced with the same dilutions of PV in suspension. Luciferase activity was determined 48 h later using the Bright-Glo™ Luciferase Assay System (#E2620, Promega, Madison, WI, USA) using a GloMax^®^ 20/20 Luminometer (Promega, Madison, WI, USA).

### 4.12. Data Analysis and Visualization

The data were analyzed and visualized using GraphPad Prism 8.0.1 Software (Boston, MA, USA).

## Figures and Tables

**Figure 1 ijms-25-12768-f001:**
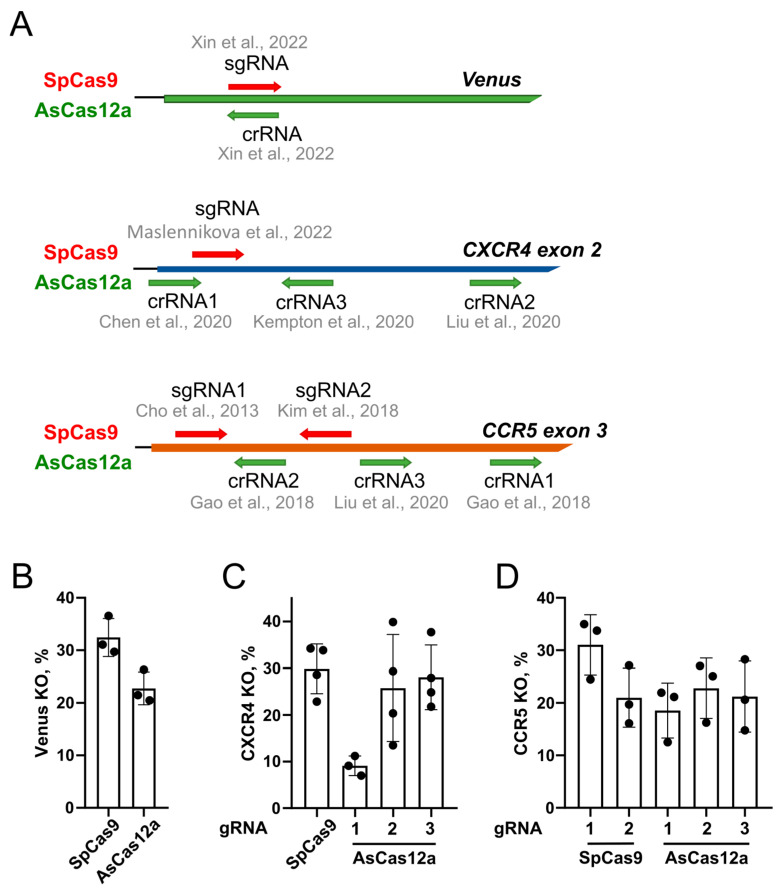
Selection of guide RNAs (gRNA). (**A**) Scheme of gRNA target sites for SpCas9 and AsCas12a in the selected loci: *Venus*, *CXCR4*, and *CCR5*; (**B**) *Venus* knockout level was measured by flow cytometry in 293-Venus clone #8 on day 6 after transfection; (**C**) *CXCR4* and (**D**) *CCR5* knockout levels were measured in CEM/CCR5 and CEM/CCR5 clone #8, respectively, stained with the corresponding antibodies on day 5 after electroporation [24,26,27,28,29,30,31,32].

**Figure 2 ijms-25-12768-f002:**
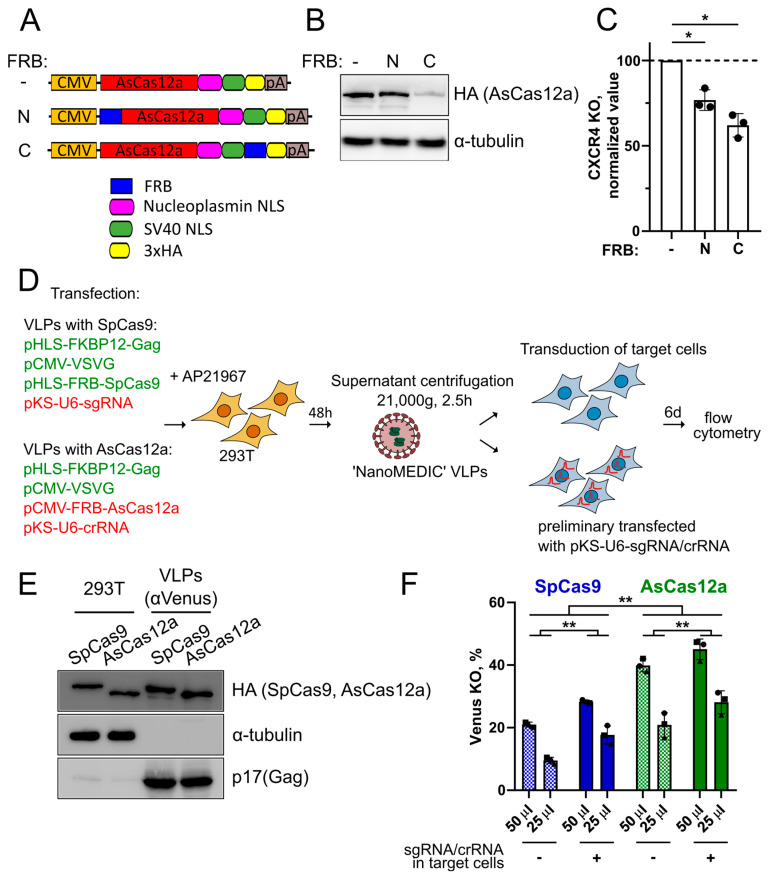
Generation of AsCas12acontaining virus-like particles (VLPs). (**A**) Scheme of plasmids encoding AsCas12a and its FRB fusion variants. (**B**) Representative Western blot evaluating the level of AsCas12a and its FRB fusion variants in transfected 293T cells. (**C**) Flow cytometry analysis of the *CXCR4* knockout level in CEM/CCR5 induced by AsCas12a and its FRB fusion variants. (**D**) Workflow of VLP production. Original plasmids used by Gee et al. to produce ‘NanoMEDIC’ particles are highlighted in green, plasmids generated and used in this study are highlighted in red. (**E**) Representative Western blot evaluating the nuclease content in lysates of 293T producer cells and VLPs targeting *Venus*. (**F**) Flow cytometry analysis of the *Venus* knockout in 293-Venus clone #8 cells mediated by VLPs with AsCas12a or SpCas9. Shaded bars correspond to target cells preliminary transfected with the plasmid encoding the corresponding gRNA, dashed bars depict target cells transfected with the plasmid coding for control gRNA. Results from three independent experiments are shown as individual data points and as mean ± standard deviation; different symbols correspond to independent experiments. Mean values were compared by (**C**) one-sample *t*-test with Bonferroni correction (*) *p* < 0.025 or (**F**) three-way ANOVA (with VLP dose, presence of crRNA in target cells, and nuclease type as factors) with subsequent Sidak’s multiple comparison test (**) *p* < 0.01.

**Figure 3 ijms-25-12768-f003:**
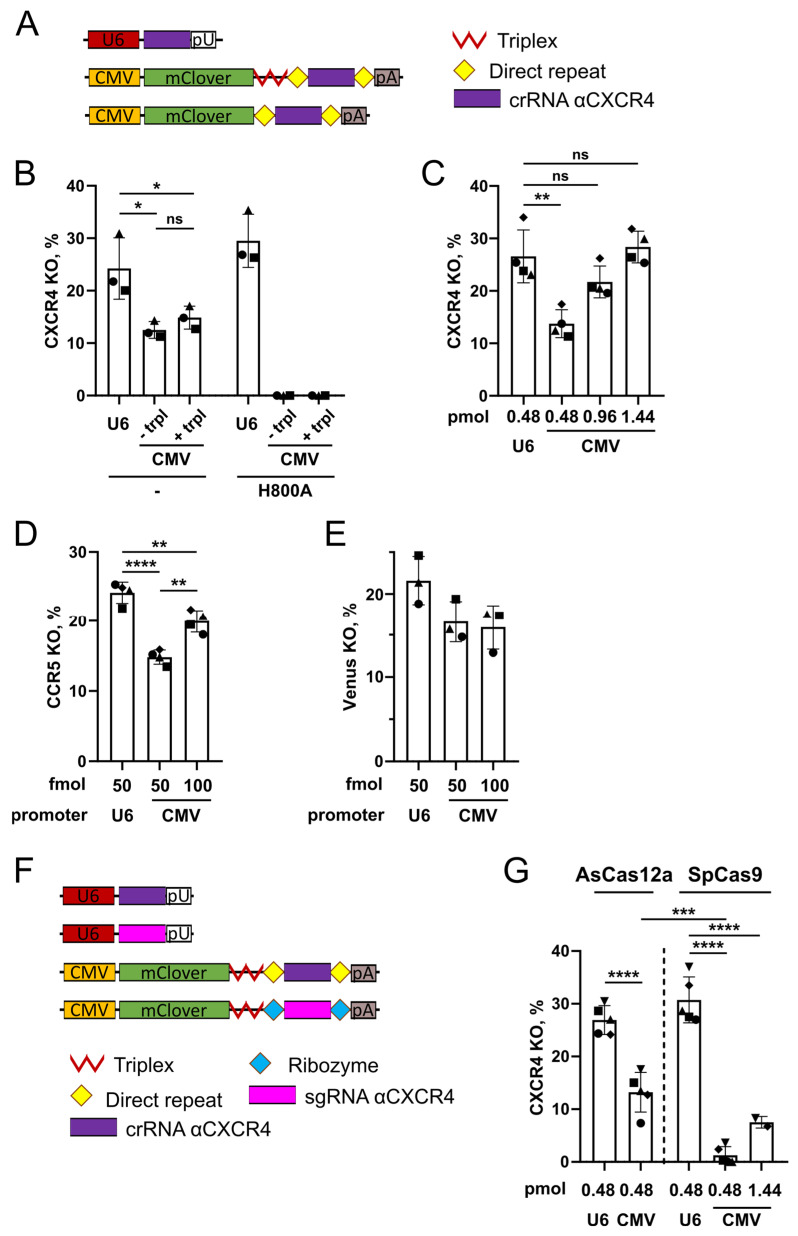
Expression of AsCas12a crRNA under the control of the RNA polymerase II (Pol II) promoter allows efficient genome editing. (**A**) Scheme of plasmids encoding crRNA under the control of the U6 or CMV promoter. (**B**) Flow cytometry analysis of the *CXCR4* knockout level in CEM/CCR5 cells mediated by AsCas12a or AsCas12a H800A in combination with one of the crRNA plasmids shown in (**A**). (**C**) Flow cytometry analysis of the *CXCR4* knockout level in CEM/CCR5 cells mediated by AsCas12a and increasing amounts of the pCMV-mClover-trpl-DR-cr1X4-DR plasmid. (**D**,**E**) Flow cytometry analysis of the *CCR5* (**D**) or *Venus* (**E**) knockout levels in 293T/CD4/CCR5 clone #19 or 293-Venus clone #8 cells, respectively, induced by AsCas12a and crRNA expressed under the control of the U6 or CMV promoter. (**F**) Scheme of plasmids encoding crRNA (AsCas12a) or sgRNA (SpCas9) under the control of the U6 or CMV promoter. (**G**) Flow cytometry analysis of the *CXCR4* knockout level in CEM/CCR5 cells mediated by AsCas12a in combination with one of the crRNA/sgRNA plasmids shown in (**F**). Results from 3–5 independent experiments are shown as individual data points and as mean ± standard deviation; different symbols correspond to independent experiments. Mean values were compared by one-way ANOVA for independent samples with subsequent Tukey’s test for multiple comparisons. (*) *p* < 0.05, (**) *p* < 0.01, (***) *p* < 0.001, (****) *p* < 0.0001, (ns)—not significant.

**Figure 4 ijms-25-12768-f004:**
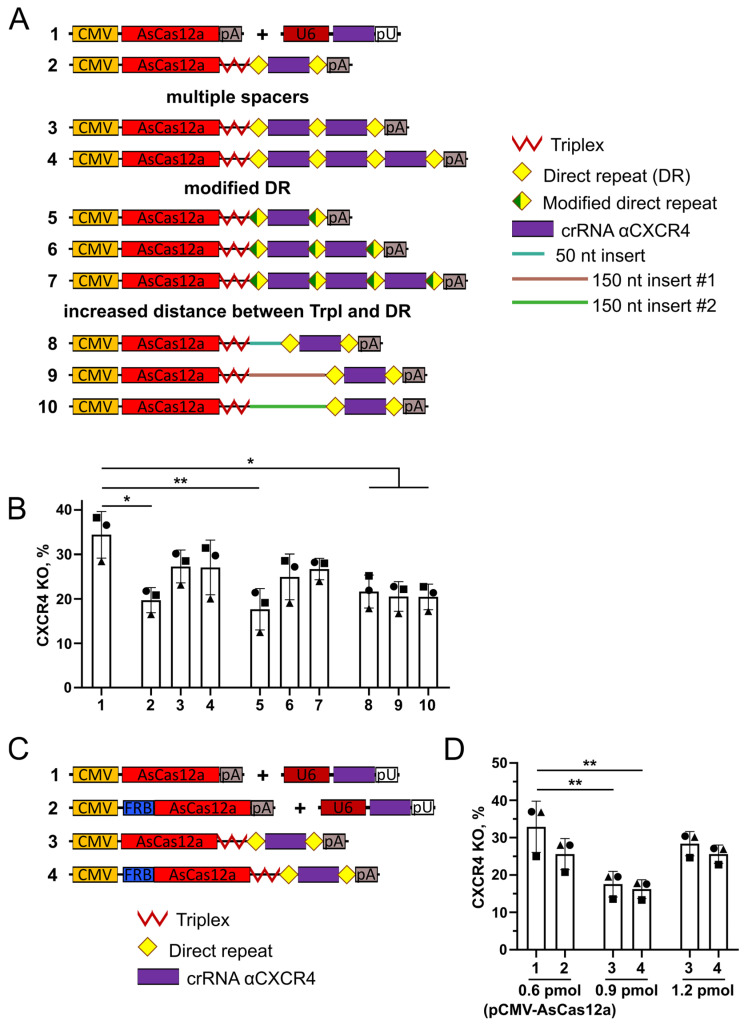
AsCas12a and crRNA can be expressed from a single Pol II-driven transcript that is compatible with the ‘NanoMEDIC’ system. (**A**) Scheme of single plasmids encoding AsCas12a and crRNA. (**B**) Flow cytometry analysis of the *CXCR4* knockout level in CEM/CCR5 cells electroporated with one of the plasmid variants shown in (**A**). (**C**) Schematic of separate and single plasmids encoding AsCas12a with or without the FRB domain. (**D**) Flow cytometry analysis of the *CXCR4* knockout level in CEM/CCR5 cells electroporated with one of the plasmid variants shown in (**C**) by AsCas12a and increasing amounts of the pCMV-mClover-trpl-DR-crRNA-DR plasmid. Results from three independent experiments are shown as individual data points and as mean ± standard deviation; different symbols correspond to independent experiments. Mean values were compared by one-way ANOVA for independent samples with subsequent Tukey’s test for multiple comparisons. (*) *p* < 0.05, (**) *p* < 0.01.

**Figure 5 ijms-25-12768-f005:**
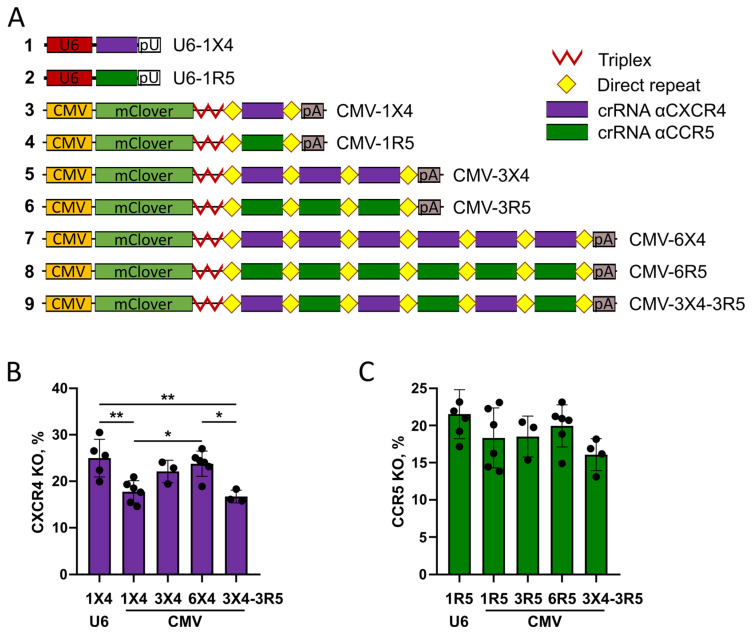
Pol II-driven crRNA is compatible with multiplex genome editing. (**A**) Scheme of plasmids encoding crRNAs. (**B**,**C**) Flow cytometry analysis of the *CXCR4* (**B**) and *CCR5* (**C**) knockout levels in CEM/CCR5 clone #8 cells electroporated with the AsCas12a plasmid together with one of the plasmids shown in (**A**). The following amounts of crRNA plasmids were used: 0.48 pmol for pKS-U6-crRNA and 0.96 pmol for pCMV-based plasmids. Results from three independent experiments are shown as individual data points as mean ± standard deviation. Mean values were compared by one-way ANOVA for independent samples with subsequent Tukey’s test for multiple comparisons. (*) *p* < 0.05, (**) *p* < 0.01 (**B**).

**Figure 6 ijms-25-12768-f006:**
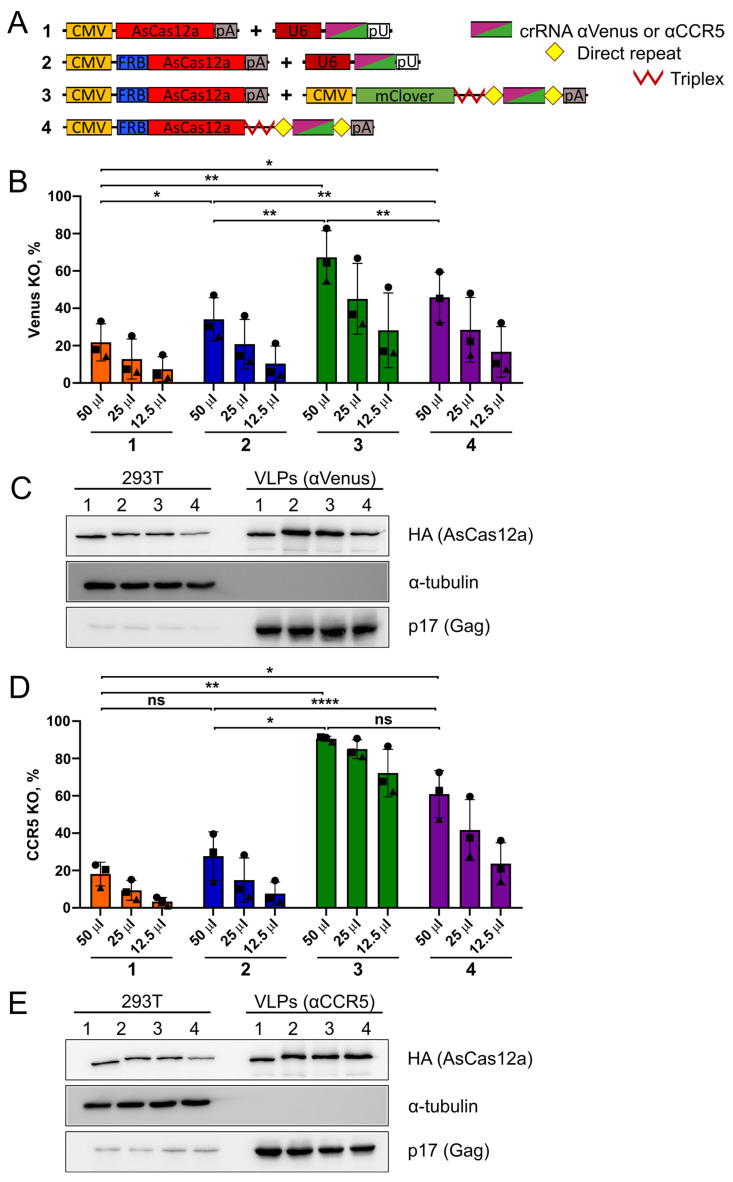
Generation of AsCas12a-VLPs with CMV promoter-driven crRNA. (**A**) Scheme of plasmids encoding AsCas12a and crRNA used for VLP production. (**B**,**D**) Flow cytometry analysis of *Venus* (**B**) and *CCR5* (**D**) knockout levels in 293-Venus clone #8 and 293T/CD4/CCR5 clone #19 cells, respectively, transduced with VLP preparations #1–4. Results from three independent experiments are shown as individual data points and as mean ± standard deviation; different symbols correspond to independent experiments. Mean values were compared by two-way ANOVA (with VLP dose and VLP type as factors) with subsequent Tukey’s multiple comparison test. (*) *p* < 0.05, (**) *p* < 0.01, (****) *p* < 0.0001, (ns)—not significant (shown only for a 50 µL dose). (**C**,**E**) Representative Western blot evaluating the nuclease content in lysates of 293T producer cells and VLPs targeting *Venus* (**C**) or *CCR5* (**E**).

**Figure 7 ijms-25-12768-f007:**
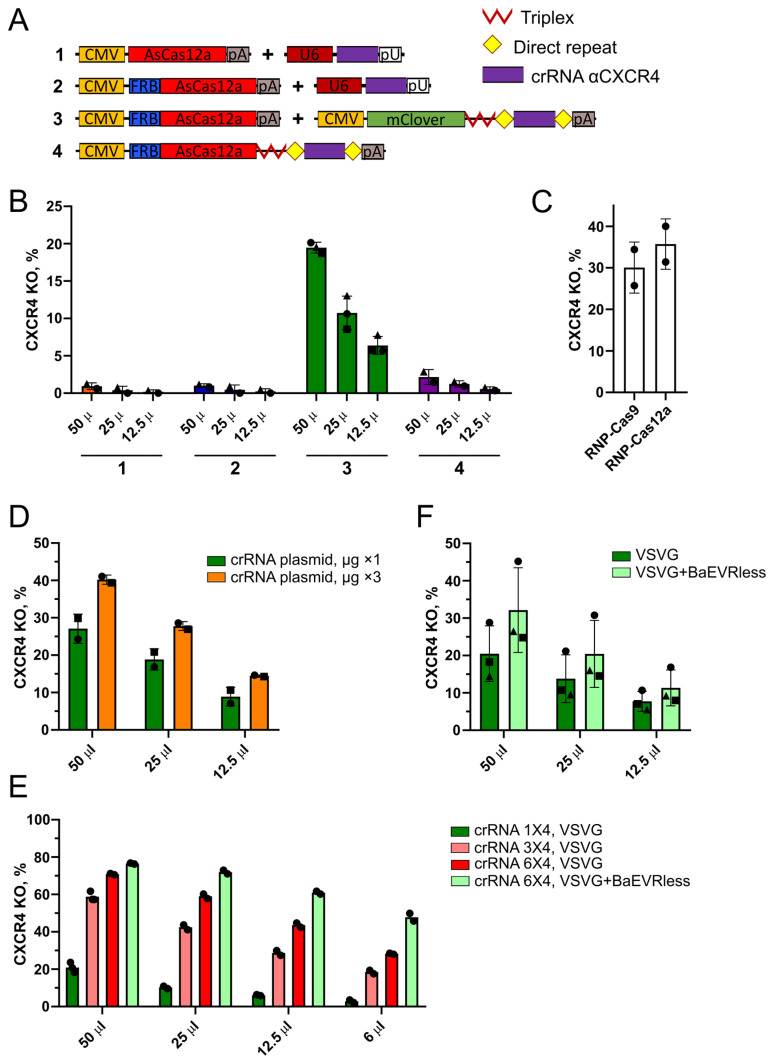
AsCas12a-VLPs produced with CMV-driven crRNA allow efficient genome editing in Jurkat T lymphocytes. (**A**) Scheme of plasmids encoding AsCas12a and crRNA used for VLP production. (**B**) Flow cytometry analysis of the *CXCR4* knockout levels in Jurkat T cells transduced with VLPs #1–4. (**C**) Flow cytometry analysis of the *CXCR4* knockout levels in Jurkat T cells electroporated with SpCas9 or AsCas12a RNPs. (**D**–**F**) Flow cytometry analysis of the *CXCR4* knockout levels in Jurkat T cells that were transduced with VLPs #3 produced with 1.66 µg or 4.98 µg of the crRNA plasmid (**D**), VLPs #3 produced with 1.66 µg of the crRNA plasmid or the plasmid coding for 3 or 6 identical spacers (**E**), and VLPs #3 coated with VSVG or VSVG+BaEVRless (data points related to independent experiments are shown by different shapes) (**F**).

## Data Availability

Data are contained within the article and Appendix A.

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
