# Peer review of "Efficient Genome Editing Using ‘NanoMEDIC’ AsCas12a-VLPs Produced with Pol II-Transcribed crRNA"

_ijms, 2024, doi:10.3390/ijms252312768_

Round 1
Reviewer 1 Report
Comments and Suggestions for Authors
The present manuscript entitled “Efficient genome editing using AsCas12a-VLPs produced with Pol II transcribed crRNA” introduces an innovative use of AsCas12a with a Pol II-driven crRNA for enhanced gene editing efficiency via VLP delivery. This approach addresses known limitations of VLP-based delivery systems, particularly the challenges with nuclear localization of RNA polymerase III-transcribed sgRNA. Overall, this study presents a promising new method for enhancing gene editing efficiency through AsCas12a-VLPs, with significant improvements over traditional SpCas9-based VLP delivery. However, there are some weaknesses that need to be addressed before this research article becomes acceptable for publication.
1. The abstract effectively presents the study's results, comparing AsCas12a-VLPs and SpCas9-VLPs, but could benefit from a sentence on the broader implications of these findings, especially potential applications for gene therapy and targeted gene editing in mammalian cells.
2. The abstract provides a clear summary of the research objective, methodology, and findings. However, the description of Pol II-driven crRNA and its role in facilitating cytoplasmic localization for VLP packaging could be expanded slightly for clarity.
3. While the introduction mentions several SpCas9-based VLP systems, it could benefit from a more detailed explanation of the limitations with SpCas9-specific VLPs, especially regarding the difficulties with gRNA packaging due to nuclear localization with the U6 promoter.
4. A short section describing AsCas12a’s mechanism, particularly its ability to excise crRNA from a longer transcript due to its RNA cleavage activity, would add clarity. The authors should explain why Pol II transcription is beneficial in VLP production.
5. The authors should describe previous attempts to improve gRNA packaging and their limitations either at introduction or at discussion.
6. At the end of introduction section, the authors could summarize the main objectives of their study, including testing the efficiency of AsCas12a with Pol II-driven crRNA in VLPs and comparing editing efficiency with SpCas9-VLPs.
7. Ensure all gRNA sequences used in the study are clearly listed in a table format, specifying target genes and positions.
8. The authors should compare efficiencies to the HEK293 and CEM/CCR5 cell results to assess system versatility.
9. It’s important to perform an off-target analysis using targeted sequencing or a high-throughput method to detect any off-target sites that may be affected by the AsCas12a-VLPs.
10. To improve the packaging efficiency of crRNA into AsCas12a-VLPs the authors could evaluate whether using different crRNA designs can increase the proportion of VLPs with crRNA-loaded AsCas12a.
11. It would be nice if the authors tracked knockout efficiency and cell viability at multiple time points post-transduction in cells to monitor cell viability.
12. Create a table summarizing the knockout efficiencies across different conditions.
Reviewer 2 Report
Comments and Suggestions for Authors
This paper describes a promising investigation on improving genome editing efficiency with AsCas12a encapsulated in NanoMEDIC virus-like particles (VLPs) containing Pol II transcribed crRNA. The study solves a major problem in CRISPR technology—efficient and targeted delivery—by combining the benefits of AsCas12a's specialised editing capabilities with a novel VLP-based method. To improve the study's scientific rigour and accessibility, various areas must be addressed, particularly defining the reason for certain methodological choices, removing redundancy within parts, and improving figure clarity.
Title
- Consider improving the title for greater specificity. While it mentions AsCas12a and VLPs, it might be more specific about the NanoMEDIC VLP system used or the innovation of coupling Pol II transcribed crRNA with AsCas12a.
- Consider rephrasing for clarity to something like: "Efficient Genome Editing Using NanoMEDIC AsCas12a VLPs with Pol II-transcribed crRNA."
Abstract
- Line 15-17: The abstract summarises the findings well, but could be more brief and emphasise the importance of AsCas12a over other CRISPR systems like SpCas9.
- Line 18: "NanoMEDIC VLPs" might be added earlier to specify the VLP platform used.
- Line 19-22: Provide context for statements about induced editing efficiency (1.5-3-fold increase) by specifying the baseline editing rate. Numerical clarity would aid readers in grasping the scope of efficiency improvements.
Introduction
- Line 29-32: The introduction provides a solid summary of CRISPR, but might better clarify the specific benefits of AsCas12a, such as its ability to edit numerous times and process crRNA. Currently, there is an imbalance, with a disproportionate focus on CRISPR/Cas9.
- Line 41-45: The book quickly presents VLP delivery method but does not explain why it is superior to other CRISPR delivery methods like lipid nanoparticles or electroporation. This is vital for readers to understand the study's context.
- Line 48-52: Emphasising AsCas12a's unique features, such as the need for a TTTV PAM site and increased specificity, would enhance the case for utilising it over SpCas9.
- Line 55-58: Although NanoMEDIC is referenced, its benefits are not fully explained. A brief explanation of how NanoMEDIC differs from standard VLPs and how it relates to genome editing might help clarify things.
Materials and Methods
- Line 210-218: The VLP production section might benefit from a discussion of optimisation processes, such as crRNA concentration and AsCas12a levels. This would improve reproducibility.
- Line 225-229: It's unclear why specific cell lines were selected for transfection. A brief justification would improve clarity, particularly for non-standard lines utilised in genome editing applications.
- Line 230-240: The study uses Pol II for crRNA transcription, but does not explain why it was chosen over Pol III for efficiency or specificity in this circumstance. This choice could be expanded to reinforce the study's methodological rationale.
- Line 250-255: Adding a graphic or flowchart to the VLP assembly and transduction cycle could help simplify difficult steps.
Results
- Line 436-456: The results section offers essential data, although there is considerable redundancy in the discussion, especially in the interpretation of CMV-driven crRNA expression outcomes. Streamlining this section and delaying interpretive comments to the discussion may improve flow.
- Line 470 reports results on Venus and CCR5 editing efficiency, however standard deviation/error numbers should be included to indicate variability. Statistical measurements in figures could help to clarify the findings.
- Lines 484-488 highlight the absence of a control group that was not modified or mocked. Without such controls, it is difficult to attribute changes primarily to the AsCas12a or VLP therapy effects. Future versions should consider incorporating these controls.
- Line 490-496: Annotations for statistical significance, such as p-values, could improve the presentation of data in figures (e.g., Figure 6a-d). Furthermore, labelling Venus and CCR5 editing percentages directly on the graphs would make these data easier to understand at a glance.
- Consider labelling experimental circumstances and cell types on Figure 2.
- Figure 6: Adding significant indicators (e.g., asterisks to represent p-values) to bar graphs can provide quick context for statistical findings. Additionally, sharper axis labels and legend descriptions would increase reader comprehension.
Discussion
- Lines 670-674: While the discussion on nuclear localisation signals (NLS) is useful, it lacks proposals for future Cas12a VLP designs. Including practical ideas for addressing problems in cytoplasmic packing would strengthen the discussion's effect.
- Line 680-690: Comparing AsCas12a VLPs to SpCas9 VLPs, especially in therapeutic applications, will help readers understand its merits. More explanation of safety and potential off-target consequences would provide more insight.
- Line 700-710: Clarify the limits of Cas12a-based systems, including potential impediments to translation to in vivo or clinical contexts.
- Lines 720-730 briefly discuss off-target effects in therapeutic circumstances. Expanding on this with references to recent studies on Cas12a specificity could strengthen the discussion.
- Line 740-750: Limitations in editing efficiency or targeting specificity should be highlighted here, along with potential future directions for optimizing VLPs for higher accuracy and reduced off-target effects.
Conclusion
- Line 776-779: The conclusion successfully summarises the study's contributions, but it may benefit from a clear overview of future research directions, such as applying AsCas12a VLPs in animal models or resolving specific difficulties in VLP delivery systems.
- Including a comment on prospective uses in clinical medicines or gene therapy would highlight the study's significance and future influence.
General Comments
- Streamline the narrative by removing redundancy between Results and Discussion sections.
- Some phrasing errors impair readability. Consider hiring a professional editor to evaluate the work for grammar, phrasing, and conciseness.
- Including confidence intervals or standard errors in graphics, particularly Figure 6, would improve statistical clarity.
- The findings could be properly contextualised by comparing them to current CRISPR and VLP investigations. This would highlight the study's uniqueness and relevance.
- For therapeutic applications, addressing safety concerns in the discussion (e.g., potential for off-target effects, immune responses) would enhance the manuscript’s rigor.
Comments on the Quality of English LanguageThe English could be improved to more clearly express the research.
Round 2
Reviewer 1 Report
Comments and Suggestions for Authors
Thank you for the opportunity to review the revised manuscript. I appreciate the authors' efforts in thoroughly addressing my suggestions. The incorporation of further analytical steps and comparisons between datasets is a crucial improvement. The emphasis on the necessity of this work for advancing biological research further underscores its relevance and importance. The revised manuscript is now comprehensive, clear, and highly informative. I believe it is suitable for publication in its current form.